# 3D-CT-GPT++: Enhancing 3D Radiology Report Generation with Direct Preference Optimization and Large Vision-Language Models

## Abstract

Automatically generating radiology reports from three-dimensional medical images, such as 3D CT scans, plays a crucial role in modern diagnostics. Current approaches for generating 3D reports often adopt video processing methods, which struggle to effectively capture the relationships along the Z-axis. Additionally, multimodal large language model-based methods for generating 3D image reports face significant limitations, particularly in terms of the image encoder's ability to represent 3D structures and the hallucinations that arise in generated content. To address these challenges, we propose the 3D-CT-GPT++ model. This model integrates the optimized 3D image encoder CTViT-V, specifically designed for chest CT scans, and builds upon the LLaVA-1.5 architecture. Furthermore, we introduce *Direct Preference Optimization (DPO)*, where GPT-4 is used to score the outputs of our fully fine-tuned (SFT) model, creating a preference dataset for subsequent DPO training. DPO significantly reduces hallucinations in the report generation process, ensuring the generated reports are more aligned with clinical needs. We fine-tuned the model on both high-quality private and public datasets to ensure clinical relevance. Extensive experiments were conducted using standard natural language generation (NLG) evaluation metrics, including BLEU, METEOR, ROUGE-L, and GREEN, to assess the report generation performance. Experimental results demonstrate that 3D-CT-GPT++ significantly outperforms existing methods in terms of accuracy, fluency, clinical factual consistency, and clinical relevance, advancing the automation of 3D medical report generation.

## 1 Introduction

Medical imaging plays a critical role in modern diagnostics, providing clinicians with precise anatomical information for accurate medical decisions (Liu et al., 2024). Three-dimensional computed tomography (3D CT), in particular, offers richer spatial information compared to two-dimensional (2D) images, aiding in the detection of pathological details that traditional techniques may miss. However, current CT image interpretation relies heavily on manual analysis by radiologists, which is time-consuming, error-prone, and adds to the clinical workload (Farahani et al., 2017). While advances have been made in generating 2D image reports (Chen et al., 2022; 2020; Qin & Song, 2022), processing 3D images, such as 3D CT scans, remains challenging due to the complexity of spatial feature extraction and high computational costs (Li et al., 2023b). Maintaining slice consistency across multiple slices is a key issue, as it is crucial for accurate diagnosis. These challenges drive the need for models that can efficiently process 3D data while preserving spatial coherence and improving diagnostic accuracy.

Despite advancements in generating reports from 3D CT images, existing approaches still face significant challenges. As shown in Figure 1, models like RadMD (Wu et al., 2023) and M3D (Bai et al., 2024) employ 3D Vision Transformers (3DViT) for feature extraction from 3D CT scans. However, processing high-dimensional 3D images with 3DViT often requires compressing high-resolution data, leading to the potential loss of critical medical details that affect diagnostic

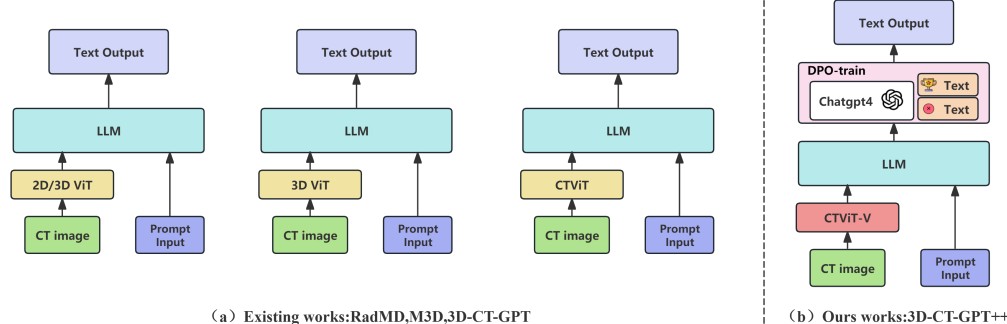

Figure 1: Comparison of architectures for RadMD, M3D-LaMed, 3D-CT-GPT, and our proposed 3D-CT-GPT++ models.

accuracy.To address these limitations, Chen et al. (2024a) introduced the 3D-CT-GPT model (see Figure 1), which integrates the CTViT encoder to enhance the extraction of spatial and temporal features. While this approach improves performance by treating 3D images as video sequences, the reliance on causal Transformers to extract temporal features limits the model's ability to fully capture dependencies across slices, especially in complex multi-slice scenarios. Additionally, the M3T (Jang & Hwang, 2022) combines CNN and Transformer for 3D image processing, but focuses on classification rather than clinically accurate radiology reports.

Moreover, a significant challenge in 3D medical report generation is the occurrence of hallucinations—where the model generates content that deviates from the actual data, resulting in inaccurate or irrelevant reports. This issue is particularly critical in the medical domain, where errors can have serious consequences. The hallucination problem is exacerbated by the lack of high-quality, aligned multimodal datasets necessary for effectively training models to produce clinically accurate reports (Liu et al., 2023b). Existing methods to mitigate hallucinations often involve Reinforcement Learning from Human Feedback (RLHF) (Li et al., 2023d), which relies on human-annotated data to guide the model's outputs. However, RLHF can be resource-intensive and challenging to scale due to the high costs and time associated with collecting human feedback.

To address these challenges, As illustrated in Figure 1(b),we optimized the original CTViT model and proposed the CTViT-V model. This model introduces a slice Transformer and relative position encoding, enhancing the feature extraction capabilities of 3D CT images, particularly in capturing global slice dependencies. Building on this improvement, we combined the LLaVA 1.5 (Liu et al., 2023a) architecture with the 3D-CT-GPT model to propose the 3D-CT-GPT++ model, which more efficiently processes 3D CT image data and generates more accurate and comprehensive radiology reports.To reduce hallucinations and avoid the scalability issues of RLHF, we adopt *Direct Preference Optimization (DPO)* (Rafailov et al., 2023), using ChatGPT-4 (OpenAI et al., 2023) to automatically score the outputs of our supervised fine-tuned (SFT) model. ChatGPT-4 (OpenAI et al., 2023) effectively mimics human judgment, providing a scalable alternative to human feedback. This approach creates a preference dataset for fine-tuning, enabling the model to generate clinically aligned reports without the high costs of manual annotations. In summary, our main contributions are:

- We propose an enhanced CTViT-V model that incorporates a slice Transformer and relative position encoding to capture global dependencies across 3D CT slices. This improvement enhances spatial coherence and diagnostic accuracy while reducing computational overhead.

- We introduce the 3D-CT-GPT++ model, based on the LLaVA-1.5 architecture, which optimizes 3D CT image processing by effectively enhancing both spatial and temporal feature extraction. This leads to more accurate and context-aware radiology reports.

- We apply *Direct Preference Optimization (DPO)* (Rafailov et al., 2023) to 3D medical imaging report generation, leveraging GPT-4 to create a preference dataset for fine-tuning.

This approach provides a practical solution to reduce hallucinations in generated reports without incurring the high costs associated with human feedback.

## 2 RELATED WORK

**Multimodal Medical Large Models (MMLMs).** In recent years, the development of Multimodal Medical Large Models (MMLMs) has greatly advanced the automatic generation of medical imaging reports. Current methods for generating medical reports mainly include model aggregation, joint vision-language model generation, and end-to-end fine-tuning approaches. Model aggregation methods combine outputs from multiple models and use specially designed prompts to generate complete reports, as seen in ChatCAD (Wang et al., 2023b) and ChatCAD+ (Zhao et al., 2024). Joint vision-language model generation methods, such as XrayGPT (Thawkar et al., 2023) and XrayPULSE, extract image features through vision encoders and integrate them with language models to achieve effective multimodal fusion. End-to-end fine-tuning methods, such as Med-PaLM (Tu et al., 2023) and XrayGLM (Wang et al., 2023a), perform joint training on image and text data, significantly improving the model's understanding and generation capabilities. Some of the leading models, such as LLaVA-Med (Li et al., 2023a), Med-PaLM2 (Singhal et al., 2023), and MedFlamingo Moor et al. (2023), have demonstrated strong performance in 2D image analysis by leveraging large-scale medical image datasets and language models. However, these models still face challenges when processing 3D images, such as CT and MRI scans, due to the complexity of spatial feature extraction and high computational costs. Although RadFM (Wu et al., 2023) and M3D-LaMed (Bai et al., 2024) have explored 3D image analysis, the generated reports still lack coherence and accuracy. Additionally, the 3D-CT-GPT (Chen et al., 2024a) model also exhibits limitations in maintaining global dependencies between slices.

**Preference Optimization and Reinforcement Learning.** Reinforcement Learning from Human Feedback (RLHF) (Li et al., 2023d) has been widely applied to improve the output quality of Large Language Models (LLMs). By incorporating human preference data, RLHF allows models to gradually learn to generate more reliable and useful outputs, especially in multimodal tasks. However, RLHF faces several challenges when applied to multimodal tasks, particularly in aligning different modalities, such as text, images, and videos. Designing an effective reward system is crucial, as poor reward design can lead to models generating inaccurate or irrelevant content. Additionally, scaling RLHF is costly, especially when collecting large-scale preference data. For instance, Sun et al. (2023) reported that collecting 10,000 human-labeled preference data points for LLaVA-RLHF cost approximately $3,000. Li et al. (2023c) also encountered scalability issues when applying GPT-4V to preference modeling, particularly when handling video inputs. Ahn et al. (2024) proposed using Supervised Fine-Tuning (SFT) models for self-evaluation, although this approach has not yet been fully validated for complex video-related tasks.

## 3 METHODOLOGY

In this section, we introduce the 3D-CT-GPT++ model, designed to enhance the automatic generation of radiology reports from 3D medical images, specifically chest CT scans. As shown in Figure 2, the model leverages the optimized 3D image encoder, CTViT-V, to efficiently process complex volumetric data while a pre-trained large language model (LLM) generates coherent and contextually accurate reports. Additionally, *Direct Preference Optimization (DPO)* (Rafailov et al., 2023) is integrated to ensure that the generated reports align with clinicians' diagnostic preferences, and feedback from ChatGPT-4 (OpenAI et al., 2023) is incorporated to further refine the reports' accuracy and coherence.

### 3.1 THE 3D ENCODER: CTVIT-V

To efficiently process 3D CT scans and capture global dependencies across slices, we propose an enhanced encoder architecture called CTViT-V, building upon the original CTViT model (Hamamci et al., 2023). Figure 2(a) illustrates the architecture of the proposed CTViT-V model. Our key improvements are:

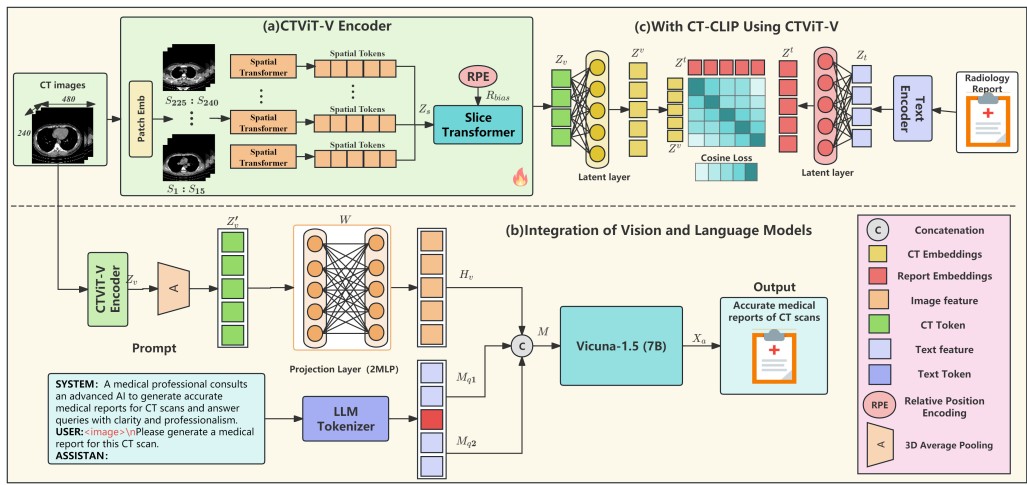

Figure 2: Architecture of 3D-CT-GPT++: The diagram illustrates how the CTViT-V encoder integrates with an LLM to generate radiology reports. CT images are processed through the Slice Transformer, and the visual features are transformed into language embeddings through an MLP, eventually generating the report.

- **Slice Transformer:** We introduce a slice Transformer module that allows for full attention across all slices, enabling the model to capture global context and dependencies in the Z-axis direction. Unlike the causal temporal attention in the original CTViT, which limits interactions to adjacent slices, our approach allows for bidirectional information flow, enhancing the model's ability to understand complex spatial relationships.

- **Relative Position Encoding:** To better model the positional relationships between slices, we incorporate relative position encoding (Shaw et al., 2018) within the slice Transformer. This allows the model to recognize the relative distances and positions of slices, improving spatial coherence and the accuracy of feature extraction.

- **Computational Efficiency:** By optimizing the attention mechanisms and incorporating efficient architectural designs inspired by ViViT (Arnab et al., 2021) and C-ViViT (Villegas et al., 2022), we reduce the computational overhead associated with processing high-resolution 3D volumes. This makes it feasible to process multi-slice CT scans without significant loss of detail.

Detailed processes are as follows:

$$Z_s = T_s(Z_x) \quad \text{where} \quad Z_x \in \mathbb{R}^{B \times S \times H \times W \times C} \tag{1}$$

Here, $Z_x$ represents the input patches of the CT volume, $T_s$ is the spatial Transformer, and $Z_s$ is the spatially encoded representation. Subsequently, the Slice Transformer, enhanced with relative position encoding, models the dependencies between the slices:

$$Z_v = T_d(Z_s + R_{\text{bias}}) \tag{2}$$

where $T_d$ is the Slice Transformer, and $R_{\text{bias}}$ represents the relative position encoding. This step allows the model to capture both local and global spatiotemporal features, resulting in improved pathological feature detection.

**Overall Encoding Process** The encoded features are then pooled using 3D average pooling to reduce both spatial and temporal resolution:

$$Z_v' = \text{AvgPool3D}(Z_v) \tag{3}$$

The pooled tensor is reshaped to flatten the spatial and temporal dimensions, producing compact feature maps suitable for integration with the language model. For detailed steps, please refer to Appendix A.1, where the full algorithm is provided.

## 3.2 VISION-LANGUAGE INTEGRATION

After obtaining the encoded visual features $Z'_v$ from the CTViT-V encoder, we integrate these features with a pre-trained large language model (LLM) to generate radiology reports. The overall architecture follows the LLaVA-1.5 (Liu et al., 2023a) framework, with modifications to accommodate the 3D visual features. As shown in Figure 2(b), we combine the pre-trained Vicuna-1.5 (7B) (Chiang et al., 2023) language model with the CTViT-V encoder. The visual features are transformed into language embedding tokens $H_v$ using a lightweight 2-layer MLP projection matrix $W$, matching the dimensionality of the word embeddings in the LLM:

$$H_v = W \cdot Z'_v. \tag{4}$$

During the report generation process, the prompt text (e.g., a clinician's question or instruction) is combined with an image placeholder to form the input prompt. This prompt is tokenized by the LLM's tokenizer, generating text tokens $M_q$. These tokens are split into $M_{q1}$ and $M_{q2}$ around the image placeholder. The visual tokens $H_v$ are concatenated with these text tokens to form the complete input:

$$M = \text{concat}([M_{q1}, H_v, M_{q2}]). \tag{5}$$

The LLM processes this input to generate the output tokens, which are decoded into the final radiology report $X_a$:

$$X_a = \text{LLM}(M). \tag{6}$$

For a detailed description of the vision-language integration and further steps involved, please refer to Appendix A.2.

## 3.3 DIRECT PREFERENCE OPTIMIZATION (DPO)

After training, we have developed a model capable of generating reports from 3D CT lung medical images. However, like other generative models, ours still encounters challenges such as hallucination, where the generated reports may include information inconsistent with real clinical scenarios. To address this, we introduce *Direct Preference Optimization (DPO)* (Rafailov et al., 2023) into 3D CT medical image report generation, drawing inspiration from LLaVA-Hound-DPO (Zhang et al., 2024). Our approach utilizes ChatGPT-4 (OpenAI et al., 2023) to score the outputs of our supervised fine-tuned (SFT) model, creating a preference dataset that guides DPO training, ensuring the generated reports align more closely with clinicians' diagnostic preferences.

### 3.3.1 CHATGPT-4 FOR SCORING AND CONSTRUCTING THE PREFERENCE DATASET

To construct the preference dataset, we first use the trained SFT model to generate a large number of medical reports from preprocessed public and private datasets. Then, these reports are evaluated using the ChatGPT-4 API. The scoring process involves inputting the ground truth reports, model-generated reports, and detailed descriptions as supporting evidence, as illustrated in Figure 3.

As shown in Figure 3(A), we sample multiple outputs from the 3D-CT-GPT++(SFT) model using a temperature setting of 1.0, ensuring diverse responses. For each 3D CT chest image and its corresponding prompt, we generate six report outputs using the 3D-CT-GPT++(SFT) model. In Figure 3(B), GPT-4 evaluates the outputs based on the evaluation prompt (*eval_prompt*), which includes both the ground truth and model-generated reports, providing feedback in the form of language-based explanations and numerical scores. The detailed evaluation prompt used for scoring is presented in Figure 4 in Appendix A.5.

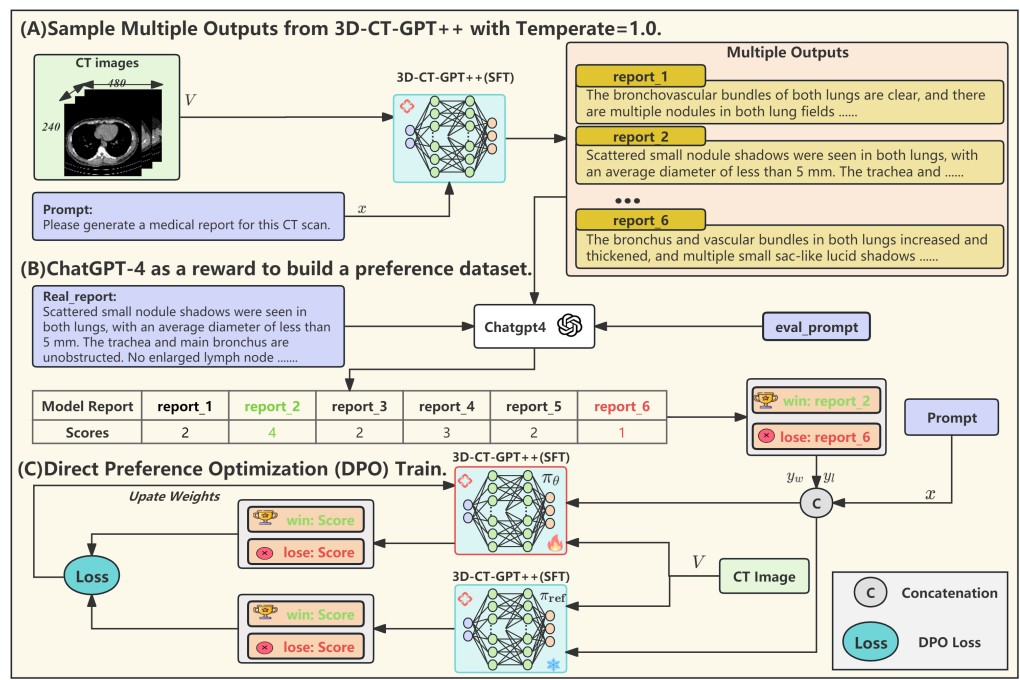

Figure 3: Overview of the Direct Preference Optimization (DPO) process for generating medical reports from 3D CT lung images. (A) Sampling diverse outputs from the model. (B) Evaluation by ChatGPT-4, comparing outputs to ground truth. (C) DPO training objective optimizing the policy model ($\pi_\theta$) against the reference model ($\pi_{\text{ref}}$).

We randomly select reports with scores $\geq 3$ as positive examples and treat reports with scores below 3 as negative examples. If all responses are uniformly scored above or below 3, the instance is excluded from the dataset. Finally, based on these scores, we construct a preference dataset. Formally, the dataset is represented as:

$$\mathcal{D}_{DPO} = \{(V, x, y_w, y_l)\} \tag{7}$$

where $V$ represents the 3D CT lung image, $x$ is the prompt, and $y_w$ and $y_l$ are the positive and negative model-generated reports, respectively.

### 3.3.2 DPO Training Objective

The *Direct Preference Optimization (DPO)* (Rafailov et al., 2023) objective is defined as follows:

$$\mathcal{L}_{\text{DPO}}(\pi_\theta; \pi_{\text{ref}}) = -\mathbb{E}_{(V, x, y_w, y_l) \sim \mathcal{D}_{\text{DPO}}}\left[ \log \sigma \left( \beta \log \frac{\pi_\theta(y_w \mid x, V)}{\pi_{\text{ref}}(y_w \mid x, V)} - \beta \log \frac{\pi_\theta(y_l \mid x, V)}{\pi_{\text{ref}}(y_l \mid x, V)} \right) \right]$$

As illustrated in Figure 3(C), $\pi_\theta$ is the policy model to be optimized, and $\pi_{\text{ref}}$ is the reference model initialized with SFT weights. Here, $\sigma$ is the logistic function, and $\beta$ is set to 0.1. This objective function optimizes the model by maximizing the log probability ratio between positive and negative samples, thereby enhancing the quality and factual consistency of the generated reports.

To clearly illustrate the *Direct Preference Optimization (DPO)* (Rafailov et al., 2023) training process, we provide detailed pseudocode in Appendix A.3.

## 3.4 DATASET

**Data Collection**   For this study, we utilized a subset of the publicly available CT-RATE dataset (Hamamci et al., 2024), which includes 25,692 non-contrast chest CT volumes. After various reconstruction techniques (Willemink & Noël, 2019), this dataset was expanded to 50,188 volumes representing 21,304 unique patients, paired with corresponding radiology reports, abnormality labels, and metadata. We directly selected the entire set of 21,304 cases from the CT-RATE dataset, with 20,000 cases allocated to the training set and 1,304 cases reserved for the testing set. Additionally, we collected 2,000 3D chest CT scans and their corresponding radiology reports from a well-known international hospital, which we refer to as *Dataset-XY*. These scans cover a wide age range (20 to 88 years) and have a mean axial resolution of 512x512 pixels, with slices per volume ranging from 100 to 600. Both datasets have been anonymized and de-identified prior to use.

**Data Preprocessing**   We performed extensive preprocessing to ensure the quality and consistency of the image and report data. For *Dataset-XY*, we applied standard de-identification protocols, removed duplicates, and filtered out irrelevant text in the reports. For the CT images, we excluded low-resolution and redundant scans, followed by manual review to further ensure consistency and uniformity. Both datasets were converted to Hounsfield Units (HU) using metadata slope and intercept values, and cropped to the range [-1000 HU, +200 HU] to reflect the diagnostic limits. Volumes were resampled to uniform spacing of 0.75 mm along the x and y axes and 1.5 mm along the z axis, and resized to a consistent resolution of 240x480x480. From the CT-RATE dataset, 20,000 cases were used for training purposes, while 1,304 cases were designated for testing to evaluate the model's performance. For a summary of dataset statistics, including the number of cases and average report length, see Table 3 in Appendix A.4.

## 3.5 TRAINING PROCESS

We divided the training process into four main stages.

**Stage 1: Image Encoder Training**   In this stage, we trained the TViT-V encoder on a large-scale 3D CT scan dataset using the CT-CLIP framework. CT-CLIP, a 3D adaptation of the CLIP architecture, was employed for self-supervised contrastive learning, aligning image and text embeddings in a shared latent space, as illustrated in Figure 2(c).

**Self-Supervised Contrastive Learning:** We utilized contrastive learning to maximize the similarity between matching 3D CT scans and corresponding radiology reports (positive pairs) while minimizing similarity between non-matching pairs (negative pairs). This method enables the model to learn effective, discriminative representations of the 3D data.

**Image Encoder Implementation:** CTViT-V extracts low-dimensional CT tokens from volumetric images, which are projected into a shared 512-dimensional space. This enables contrastive learning with the text encoder, ensuring robust image representation.

**Text Encoder Integration:** For the text encoder, we used a pre-trained CXR-BERT, which encodes radiology reports into the same 512-dimensional space for contrastive alignment.

**Training Objective:** The primary training objective is to align image and text embeddings by maximizing the cosine similarity between positive pairs and minimizing the similarity between negative pairs. This optimizes the CTViT-V encoder for extracting features from 3D CT images, ensuring effective medical report generation.

**Stage 2: Pre-training**   In this stage, the model was trained to understand the relationship between 3D CT image features and their corresponding reports by analyzing a large set of 3D CT image-report pairs. During this process, both the image encoder and language model were frozen, and we focused on training the projection layer. The training was conducted using a custom-built dataset, which comprises both public and private datasets. Due to the scarcity of paired 3D CT images and reports, we were unable to perform the large-scale alignment training typical of multimodal models. Instead, we adopted a pre-training approach by training separately on the public dataset and the private dataset, and comparing the effectiveness of different methods to address this challenge.

**Stage 3: Fine-tuning** In this stage, we further refined the model to align 3D CT image features with specific radiology reports. We employed two fine-tuning strategies ,where the fine-tuning approach resembles that described in LLM-CXR (Lee et al., 2024).

**LoRA-based Lightweight Fine-tuning** During this phase, we employed LoRA (Low-Rank Adaptation) (Hu et al., 2021) for lightweight fine-tuning. The image encoder remained frozen, while parts of the language model and projection layers were fine-tuned. This efficient approach helped avoid overfitting.

**Supervised Fine-tuning (SFT)** Additionally, during this phase, we applied supervised fine-tuning (SFT) (Brown et al., 2020), optimizing all parameters of the language model using labeled data. This contrasts with parameter-efficient methods like LoRA, which adjust only a subset of layers. The number of training epochs was carefully selected through cross-validation to prevent overfitting.

**Stage 4: Direct Preference Optimization (DPO)** After fine-tuning, the model underwent *Direct Preference Optimization (DPO)* to ensure that generated reports aligned more closely with clinicians' diagnostic preferences.

**GPT-4 Scoring and Preference Dataset** GPT-4 was used to score the generated reports based on factual consistency and coherence, producing a preference dataset. Reports with scores above a threshold were used as positive examples, and those below the threshold served as negative examples, as described in Section 3.3.1.

**DPO Training** Using this preference dataset, the model was trained to optimize report generation by adjusting the likelihood of producing higher-scoring reports, following the DPO objective. This helped the model generate more clinically relevant and accurate reports,as described in Section 3.3.2.

## 4 EXPERIMENTS

In this section, we present the experimental setup, results, and analysis to evaluate the performance of our proposed model, 3D-CT-GPT++.

### 4.1 EXPERIMENTAL SETUP

Our experiments consist of four main stages: Image Encoder Training, Model Pre-training, Fine-tuning, and Direct Preference Optimization (DPO). Detailed hardware configurations, hyperparameter settings, and implementation specifics are provided in Appendix B.1. To ensure robustness and stability, we ran five experiments for each model variant and computed the average scores, minimizing the influence of random factors and ensuring more reliable performance evaluations. This methodology was consistently applied across all experiments. The evaluation metrics used in our analysis are detailed in Appendix B.2. Additionally, for each model configuration, specific training details—such as dataset preparation, number of training epochs, batch sizes, and optimizer settings—are provided in Appendix B.3 to ensure transparency and reproducibility. All experiments were conducted with a temperature setting of 0.7, except for those in the subsubsection 4.2.3.

### 4.2 RESULTS AND ANALYSIS

Table 1: Overall Performance and GREEN Scores Comparison of 3D-CT-GPT and Variants. The table presents BLEU-1, BLEU-4, ROUGE-1, ROUGE-2, ROUGE-L, METEOR scores, and GREEN scores. The best and second-best results for each metric are highlighted in **bold** and underlined, respectively.

| Model / Method | BLEU-1 | BLEU-4 | ROUGE-1 | ROUGE-2 | ROUGE-L | METEOR | GREEN |
|---|---|---|---|---|---|---|---|
| **(A) Overall Performance** | | | | | | | |
| 3D-CT-GPT++ (LoRA) | 55.98 | 10.50 | 0.4561 | 0.2209 | 0.3306 | 0.3061 | 0.2546 |
| 3D-CT-GPT++ (SFT) | 54.65 | 10.16 | 0.4505 | 0.2123 | 0.3199 | 0.2995 | 0.2596 |
| 3D-CT-GPT++ (SFT+DPO) | **56.76** | **13.32** | **0.5117** | **0.2467** | **0.3692** | **0.3542** | **0.3527** |

### 4.2.1 OVERALL PERFORMANCE

As shown in Section (A) of Table 1, our model 3D-CT-GPT++ (SFT+DPO) achieves the best performance across all metrics compared to its counterparts. It significantly outperforms 3D-CT-GPT++ (LoRA) and 3D-CT-GPT++ (SFT). For instance, BLEU-4 improves from 10.50 and 10.16 in the LoRA and SFT variants to **13.32** with our model, indicating enhanced accuracy and coherence in generated reports. Similarly, ROUGE-L increases from 0.3306 and 0.3199 to **0.3692**, suggesting that our model produces reports with structures more closely aligned with reference texts. In particular, the GREEN score for 3D-CT-GPT++ (SFT+DPO) reaches the highest value of **0.3527**, indicating superior overall performance in terms of both content and coherence. The improvements across all metrics reflect the effectiveness of our approach, integrating SFT with DPO for enhanced fine-tuning and preference optimization.

### 4.2.2 COMPARISON WITH EXISTING MODELS

As shown in Section (B) of Table 2, our model 3D-CT-GPT++ significantly outperforms existing models RadFM and M3D (from the literature), as well as our baseline 3D-CT-GPT, across all comparable metrics. For example, our model achieves a ROUGE-L score of 0.3692, surpassing RadFM's 15.51, M3D's 19.55, and the baseline's 0.3353. The METEOR score also improves from the baseline's 0.3308 to 0.3542, notably higher than M3D's 0.1438. These results highlight the effectiveness of our approach, attributed to the advanced encoder architecture and the incorporation of DPO, which better align the model's outputs with human preferences.

### 4.2.3 IMPACT OF TEMPERATURE ON PERFORMANCE

Section (C) of Table 2 explores the effect of varying the temperature parameter during inference on the performance of 3D-CT-GPT++. A temperature of 0.4 yields the highest scores for most metrics, including BLEU-1, BLEU-4, ROUGE-1, and METEOR. Specifically, at temperature 0.4, the model achieves a BLEU-4 score of 14.47 and a ROUGE-L score of 0.3807. However, higher temperatures (e.g., 0.8 and 0.9) lead to a slight decline in performance metrics. This is expected, as higher temperatures introduce more randomness, increasing the diversity of the generated text but potentially reducing the overlap with reference texts, as measured by BLEU and ROUGE scores. Based on these observations, a temperature of 0.4 appears to provide the best balance between diversity and accuracy for our task. However, depending on the specific requirements of report generation—such as the need for more deterministic outputs—a temperature of 0.7 may still be preferable.

### 4.2.4 IMPACT OF DATA QUANTITY ON DPO PERFORMANCE

Section (D) of Table 2 compares the supervised fine-tuned model 3D-CT-GPT++ (SFT), the DPO-trained model using the initial data selection method 3D-CT-GPT++ (1), and the DPO-trained model with a refined data selection strategy 3D-CT-GPT++ (2). The refined strategy in 3D-CT-GPT++ (2) aimed to increase contrast by selecting the highest-scoring candidate as the positive example and the lowest-scoring as the negative; if no candidate scored $\geq 3$, we used the real report as the positive example. However, this refined strategy resulted in decreased performance compared to 3D-CT-GPT++ (1); for example, BLEU-4 decreased from 13.32 to 12.41. Similarly, Chen et al. (2024b) introduced a self-play method for DPO training, preferring ground-truth cases over model-generated responses. Zhang et al. (2024) found that their LLaVA-Hound-DPO (Zhang et al., 2024) model showed a 3% accuracy decline compared to SFT models when using self-play (Chen et al., 2024b). These findings suggest that while reward incorporation benefits complex tasks, extreme contrasts between examples may hinder performance. Our results support this, indicating that in DPO training, the quality and representativeness of preference data are more important than the quantity or extremity of examples.

Table 2: Performance comparison of 3D-CT-GPT and M3D across different training strategies and datasets. The table presents the evaluation metrics BLEU-1, BLEU-4, ROUGE-1, ROUGE-2, ROUGE-L, and METEOR for different models, training strategies, and datasets. The best and second-best results for each metric are highlighted in **bold** and underlined, respectively. Sections are marked with labels (B) to (D) for easy reference in the text.

| Model / Method | BLEU-1 | BLEU-4 | ROUGE-1 | ROUGE-2 | ROUGE-L | METEOR |
|---|---|---|---|---|---|---|
| **(B) Comparison with Existing Models** | | | | | | |
| RadFM (Literature Results) | 10.21 | - | - | - | 0.1551 | - |
| M3D (Literature Results) | 15.15 | - | - | - | 0.1955 | 0.1438 |
| 3D-CT-GPT (Baseline) | 52.17 | 11.49 | 0.4711 | 0.2224 | 0.3353 | 0.3308 |
| 3D-CT-GPT++ | **56.76** | **13.32** | **0.5117** | **0.2467** | **0.3692** | **0.3542** |
| **(C) Impact of Temperature** | | | | | | |
| Temperature 0.4 | **57.87** | **14.47** | **0.5327** | **0.2593** | **0.3807** | **0.3695** |
| Temperature 0.5 | 57.12 | 14.13 | 0.5250 | 0.2559 | 0.3784 | 0.3651 |
| Temperature 0.6 | 56.79 | 13.48 | 0.5148 | 0.2466 | 0.3705 | 0.3559 |
| Temperature 0.7 | 56.76 | 13.32 | 0.5117 | 0.2467 | 0.3692 | 0.3542 |
| Temperature 0.8 | 56.93 | 13.42 | 0.5068 | 0.2463 | 0.3675 | 0.3486 |
| Temperature 0.9 | 55.61 | 13.16 | 0.5013 | 0.2415 | 0.3616 | 0.3494 |
| **(D) Impact of Data Quantity on DPO Performance** | | | | | | |
| 3D-CT-GPT++ (1) | **56.76** | **13.32** | **0.5117** | **0.2467** | **0.3692** | **0.3542** |
| 3D-CT-GPT++ (2) | 55.38 | 12.41 | 0.4857 | 0.2309 | 0.3479 | 0.3339 |
| 3D-CT-GPT++ (SFT) | 54.65 | 10.16 | 0.4505 | 0.2123 | 0.3199 | 0.2995 |

## 4.3 QUALITATIVE ANALYSIS

We compare generated reports from various versions of 3D-CT-GPT++ (DPO, SFT, LoRA, and the baseline model) with the real medical report. Examples are provided in Figure 5 of Appendix D. The 3D-CT-GPT++ models consistently produce more accurate and detailed reports, capturing clinical findings and using appropriate medical terminology. Compared to baseline models, 3D-CT-GPT++ shows clear improvements in report quality and accuracy.

## 5 CONCLUSION

We have introduced 3D-CT-GPT++, a novel model for radiology report generation that leverages advanced encoder architectures and Direct Preference Optimization to achieve superior performance. Through comprehensive experiments and ablation studies, we demonstrate the model's effectiveness and potential for clinical applications.

### 5.1 LIMITATIONS AND FUTURE WORK

While 3D-CT-GPT++ has demonstrated significant advancements in radiology report generation, several limitations remain. Improving the clinical relevance of generated reports requires integrating patient-specific information, such as medical history and symptoms. Future work will explore combining clinical background data with imaging features to produce reports better aligned with real-world needs. We also plan to expand our dataset to include diverse medical imaging types, such as X-rays and MRIs, to enhance generalization. Collaborating with medical institutions, we aim to conduct large-scale clinical trials to gather clinician feedback and optimize model performance in real-world settings. Additionally, we will develop evaluation metrics focused on clinical relevance and improve model interpretability to ensure accuracy and clarity for clinicians. To address computational challenges, we will explore more efficient model architectures and optimization techniques to reduce training and inference costs, facilitating deployment in resource-constrained healthcare environments. By addressing these limitations and pursuing these directions, we aim to enhance the performance and practicality of 3D-CT-GPT++, promoting its adoption in clinical radiology.

REPRODUCIBILITY STATEMENT

We have made significant efforts to ensure the reproducibility of our results. All details regarding the architecture of 3D-CT-GPT++, including model parameters, training settings, and hyperparameter configurations, are clearly documented in the main text and Appendix.

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

# A   ALGORITHM

## A.1   3D CT IMAGE ENCODER ALGORITHM

This algorithm describes the process of encoding 3D CT images into spatial-temporal feature representations, which is a key step in extracting meaningful visual information for report generation.

---

**Algorithm 1:** 3D CT Image Encoding Process

---

**Input:** Normalized 3D CT image $x \in \mathbb{R}^{B \times 1 \times 240 \times 480 \times 480}$, where $B$ is the batch size and 240 is the normalized number of slices.

**Output:** Encoded features $Z'_v \in \mathbb{R}^{B \times (8 \times 8 \times 8) \times 512}$, where $(8 \times 8 \times 8)$ are the reduced dimensions after pooling and 512 is the embedding dimension for each patch.

1 **Step 1: Patch Segmentation and Embedding**
2 Divide the 3D CT image $x$ into non-overlapping patches of size $15 \times 30 \times 30$:
3    $Z_x \leftarrow \text{PatchEmbedding}(x)$   where   $Z_x \in \mathbb{R}^{B \times 16 \times 16 \times 16 \times 512}$
4 **Step 2: Spatial Encoding**
5 Apply the spatial transformer $T_s$ to each patch:
6 **for** $s = 1$ **to** 16 **do**
7    $\lfloor$   $Z_s \leftarrow T_s(Z_x)$ // Encode spatial dependencies
8 Stack the encoded patches:
9 $Z_s \in \mathbb{R}^{B \times 16 \times 16 \times 16 \times 512}$
10 **Step 3: Slice Encoding**
11 Apply relative position bias $R_{bias}$ and use slice transformer $T_d$:
12 **for** $z_i \in Z_s$ **do**
13    $\mid$   $A_{bias\_slice} \leftarrow R_{bias}(16)$
14    $\lfloor$   $Z_v \leftarrow T_d(Z_s, A_{bias\_slice})$ // Encode slice-wise dependencies
15 After slice encoding:
16 $Z_v \in \mathbb{R}^{B \times 16 \times 16 \times 16 \times 512}$
17 **Step 4: Apply 3D Average Pooling**
18 Apply 3D average pooling with kernel size $2 \times 2 \times 2$:
19 $Z_v \leftarrow \text{AvgPool3D}(Z_v, \text{kernel size} = 2)$   where   $Z_v \in \mathbb{R}^{B \times 8 \times 8 \times 8 \times 512}$
20 **Step 5: Reshape the Tensor**
21 Reshape pooled tensor to merge dimensions:
22 $Z'_v \leftarrow \text{Reshape}(Z_v, [B, (8 \times 8 \times 8), 512])$   where   $Z'_v \in \mathbb{R}^{B \times (8 \times 8 \times 8) \times 512}$
23 **Step 6: Return Projected CT Tokens**
24 **return** $Z'_v$

---

## A.2   VISUAL-LANGUAGE INTEGRATION PROCESS

Once the 3D CT images are encoded into feature representations, the next step is to integrate these features with a pre-trained large language model (LLM). This process is outlined in the following algorithm, which corresponds to the integration and report generation components discussed in Section **3.2** of the main paper.

---

**Algorithm 2:** Visual-Language Integration Process (3D-CT-GPT++)

---

**Input:** Normalized 3D CT image $X_v \in \mathbb{R}^{B \times C \times S \times H \times W}$, Query text $Q$
**Output:** Generated report $X_a$

1 **1: Extract Visual Features**
2 $Z_v \leftarrow g(X_v)$ // Extract visual features using trained CTViT encoder (Algorithm 1)
3 $H_v \leftarrow W(Z_v)$ // Project visual features into language embedding space using 2-layer MLP
4 **2: Process Query Text**
5 $Q_{tokens} \leftarrow \text{LLM\_Tokenizer}(Q)$ // Tokenize the query text
6 $M_q \leftarrow \text{LLM\_Embedding}(Q_{tokens})$ // Map tokens into word embeddings
7 $M_{q1}, M_{q2} \leftarrow \text{Split}(M_q)$ // Split text embedding into two parts around image placeholder
8 **3: Concatenate Text and Visual Features**
9 $M \leftarrow \text{Concat}([M_{q1}, H_v, M_{q2}])$ // Concatenate visual and text embeddings
10 **4: Generate Output using LLM**
11 $X_{a\_tokens} \leftarrow g(M)$ // Pass the combined embeddings into the LLM to generate output tokens
12 $X_a \leftarrow \text{LLM\_Decoder}(X_{a\_tokens})$ // Decode output tokens into final report
13 **5: Return Generated Report**
14 **return** $X_a$

---

## A.3 DIRECT PREFERENCE OPTIMIZATION (DPO)

---

**Algorithm 3:** Direct Preference Optimization (DPO) Training Algorithm

---

**Input :** Pretrained model $\pi_\theta$, Reference model $\pi_{\text{ref}}$, GPT-4 scoring function, Dataset $\mathcal{D}_{\text{DPO}}$, Learning rate $\alpha$, Number of epochs $T$, Temperature parameter $\tau$, Scaling parameter $\beta = 0.1$
**Output:** Optimized model $\pi_\theta$

1 **for** *each epoch $t = 1$ to $T$* **do**
2    **for** *each data sample $(V, x) \in \mathcal{D}_{DPO}$* **do**
3       Generate multiple outputs $y_1, \ldots, y_6$ from model $\pi_\theta$;
4       Score each output using GPT-4: $s_i = \text{GPT-4\_score}(y_i)$ for $i = 1, \ldots, 6$;
5       Select $y_w$ where $s_w \geq 3$ as the positive example;
6       Select $y_l$ where $s_l < 3$ as the negative example;
7       **if** *no $y_w$ with $s_w \geq 3$ or no $y_l$ with $s_l < 3$* **then**
8          **continue** to next sample;
9       **else**
10          Compute $\Delta = \beta \left( \log \frac{\pi_\theta(y_w|x,V)}{\pi_{\text{ref}}(y_w|x,V)} - \log \frac{\pi_\theta(y_l|x,V)}{\pi_{\text{ref}}(y_l|x,V)} \right)$;
11          Compute DPO loss: $\mathcal{L}_{\text{DPO}} = -\log \sigma(\Delta)$;
12          Update model parameters: $\theta \leftarrow \theta - \alpha \nabla_\theta \mathcal{L}_{\text{DPO}}$;
13       **end**
14    **end**
15 **end**
16 **return** Optimized model $\pi_\theta$;

---

## A.4 DATASET STATISTICS

Table 3: Dataset statistics for CT-RATE and Dataset-XY.

| Dataset | CT-RATE | | | Dataset-XY | | |
|---|---|---|---|---|---|---|
| | **Train** | **Test** | **Val** | **Train** | **Test** | **Val** |
| **Images** | 17000 | 652 | 652 | 1508 | 190 | 188 |
| **Reports** | 17000 | 652 | 652 | 1508 | 190 | 188 |
| **Avg. Length (words)** | 198.5 | 197.5 | 198.8 | 88.4 | 88.6 | 88.9 |

For the publicly collected dataset, we initially had 20,000 samples designated for training. After preprocessing, we retained 17,000 samples for the training set. Additionally, the original validation set of 1,304 samples was split evenly into 652 samples for testing and 652 samples for validation. This results in the dataset distribution presented in Table 3, where CT-RATE comprises 17,000 training samples, 652 testing samples, and 652 validation samples. Similarly, Dataset-XY includes 1,508 training samples, 190 testing samples, and 188 validation samples. The average length of reports varies between the datasets, with CT-RATE reports averaging approximately 198 words and Dataset-XY reports averaging around 88 words.

## A.5 DETAILED EVALUATION PROMPT

Your task is to act as an impartial and objective assessor of CT reports. Utilizing the ground truth and the model-generated CT report, your evaluation should focus on the following aspects:

- **Accuracy**: Does the model-generated report accurately reflect the findings and diagnosis in the real CT report?

- **Completeness**: Does the model-generated report cover all key details and findings? Does it omit any important information?

- **Clarity**: Is the model-generated report clear? Are there any repetitions, unclear descriptions, or grammatical errors?

- **Consistency**: Is the model-generated report consistent with the conclusions and descriptions in the real report?

**Input**:
Real CT Report: {real_report}
Model-Generated Report: {model_report}

**Output Format**:
Explanation: <brief judgement of prediction>
Score: <an integer score of quality from 1-5>

**Samples:**
**Input**:

Real CT Report: {The bronchovascular bundles of both lungs are clear, no obvious major lesions are seen, the bronchus of each lobe and segment are unobstructed, no enlarged lymph nodes are seen in the mediastinum and hilus of both lungs, no obvious abnormalities are seen in the heart and major blood vessels, no thickening of bilateral pleura, and no effusion is seen in bilateral thoracic cavity. Multiple slightly larger lymph nodes were seen in both armpits, with the largest one having a short diameter of 8 mm.}
Model-Generated Report: {The bronchovascular bundles of both lungs are clear, no obvious major lesions are seen, the bronchus of each lobe and segment are unobstructed, a few cord shadows are visible in the lower lobes of the left and right lung, multiple small lymph nodes are scattered in both hilars, with a short diameter of less than 5 mm, some aorta calcification is visible, and no significant effusion is seen in bilateral thoracic cavity.}

**Output **:
Explanation: <The model-generated report is partially accurate, but it includes additional findings not mentioned in the real CT report, such as cord shadows in the lower lobes of the lungs, scattered small lymph nodes in both hilars, and aorta calcification. These discrepancies affect the accuracy and consistency of the report. The report is clear and grammatically correct, but it is not complete as it omits the finding of multiple slightly larger lymph nodes in both armpits from the real report.>
Score: <3/5>

Figure 4: Detailed evaluation prompt used by GPT-4 for assessing radiology reports. This prompt includes both the ground truth and the model-generated reports, along with instructions for providing feedback and scoring.

## B  DETAILED EXPERIMENTAL SETUP

### B.1  HARDWARE CONFIGURATION, HYPERPARAMETER SETTINGS, AND IMPLEMENTATION DETAILS

During the development of our model, we conducted extensive experiments on various GPU configurations to ensure efficiency and optimization across different training phases. The training was conducted on a **private dataset**, and the corresponding GPU configurations, hyperparameters, and resource utilization are summarized below.

For the **image encoder training phase**, we utilized an L20 GPU with 48GB of memory, supporting a batch size of 4, which occupied approximately 46GB of GPU memory. The learning rate was set to $1.25 \times 10^{-6}$, ensuring stable and efficient training.

In the **pre-training phase**, we employed a single RTX 3090 GPU (24GB memory). With a batch size of 1, this phase required approximately 14GB of GPU memory, and a learning rate of $1 \times 10^{-3}$ was applied.

For the **LoRA-based fine-tuning phase**, the learning rate was adjusted to $2 \times 10^{-4}$ while maintaining a batch size of 1. The memory usage during this phase increased to approximately 22GB. The Adam optimizer with a cosine learning rate scheduler and bfloat16 precision was utilized to enhance computational efficiency. The **SFT phase** used a single NVIDIA A100 GPU (80GB memory), with each batch occupying around 68GB of GPU memory. In the **DPO fine-tuning phase**, we adopted a hybrid training strategy. The GPU memory utilization was approximately 28GB, while 30 CPU cores were engaged for computation. A learning rate of $5 \times 10^{-7}$ was used during this stage. The key hyperparameters for these training phases are summarized in Table 4.

Table 4: Key hyperparameters for training on the private dataset.

| Hyperparameter | Pre-training | LoRA Fine-tuning | SFT Fine-tuning | DPO Fine-tuning |
|---|---|---|---|---|
| Learning Rate | $1 \times 10^{-3}$ | $2 \times 10^{-4}$ | $2 \times 10^{-5}$ | $5 \times 10^{-7}$ |
| Scheduler | Cosine | Cosine | Cosine | Linear |
| Warmup Ratio | 0.03 | 0.03 | 0.03 | 0.1 |
| Batch Size | 1 | 1 | 1 | 1 |
| Epochs | 5 | 2 | 2 | 1 |
| Weight Decay | 0.0 | 0.0 | 0.0 | 0.0 |
| Dropout Rate | 0.1 | 0.1 | 0.1 | 0.1 |
| Hidden Size | 512 | 512 | 512 | 512 |
| Max Model Length | 2048 | 2048 | 2048 | 2048 |
| Lazy Preprocess | True | True | True | True |
| Save Strategy | Steps | Steps | Steps | Steps |
| Special Settings | - | LoRA ($r = 128, \alpha = 256$) | Pretrained Adapter | Freeze MLP Adapter |

### B.2  EVALUATION METRICS

To evaluate our radiology report generation model, we use standard NLG metrics: BLEU (Papineni et al., 2002), METEOR (Banerjee & Lavie, 2005), and ROUGE-1, ROUGE-2, and ROUGE-L (Lin, 2004), computed using the sacrebleu (Post, 2018), nltk, and rouge_score libraries. BLEU measures n-gram overlap up to four in length, focusing on precision, with scores ranging from 0 to 100. In contrast, METEOR and ROUGE scores typically range from 0 to 1. METEOR accounts for synonyms, stemming, and paraphrasing, balancing precision and recall to capture both accuracy and completeness. ROUGE-1, ROUGE-2, and ROUGE-L measure unigram, bigram, and longest common subsequence overlap, respectively, assessing fluency and coherence. These metrics collectively provide a comprehensive evaluation of text quality, focusing on precision, recall, and structural similarity. Additionally, to assess clinical accuracy and factual correctness, we employ the GREEN metric (Ostmeier et al., 2024), which is specifically designed for evaluating radiology reports by detecting factual errors and hallucinations. By combining traditional NLG metrics with the GREEN metric, we ensure a comprehensive evaluation of both linguistic quality and clinical relevance of the generated reports. While they don't directly measure clinical accuracy, they are widely accepted for assessing the linguistic quality of generated reports in medical applications.

### B.3    Training Details for Result and Ablation

### B.3.1    Overall Performance Experimental Setup

To evaluate the performance of different versions of 3D-CT-GPT++ (LoRA, SFT, and SFT+DPO), we used the private Dataset-XY, with all models utilizing the CTViT-V image encoder. The training details are as follows: The image encoder was trained on Dataset-XY (train) for 6000 steps, divided into 4 batches, taking approximately 3.3 hours. For the pre-training phase, we conducted 5 epochs of single-batch training on Dataset-XY (train), lasting around 1.3 hours. In the fine-tuning phase, both LoRA and SFT models were fine-tuned on Dataset-XY (train) for 2 epochs with a single batch. LoRA training required approximately 1.4 hours, while SFT took about 3.6 hours, resulting in the 3D-CT-GPT++ LoRA and SFT versions. Lastly, the DPO phase, based on the SFT model, was trained using a preference dataset generated from Dataset-XY (test), taking approximately 2.4 hours and yielding the final version, 3D-CT-GPT++ (SFT+DPO).

### B.3.2    Comparison with Existing Models

In the *Comparison with Existing Models* experiment, we compared 3D-CT-GPT (baseline), 3D-CT-GPT++, RadFM, and M3D. For 3D-CT-GPT, based on the architecture from Chen et al. (2024a), we first trained the CTViT encoder using Dataset-XY (train) for 6,000 steps, followed by 5 epochs of pre-training and 2 epochs of LoRA fine-tuning, adhering strictly to the original 3D-CT-GPT model structure. For 3D-CT-GPT++, we used the model form described in Appendix B.3.1, specifically the 3D-CT-GPT++ (SFT+DPO) version as the final form. Although RadFM supports both 2D and 3D inputs, we were unable to fine-tune the model on our dataset due to time and resource constraints. Fine-tuning RadFM would require additional adaptations, as its data processing format differs from ours. Similarly, M3D, as a multi-task model, requires segmentation as part of its pipeline, which involves significant pre-processing and annotated segmentation data, exceeding the scope of this work. Furthermore, M3D's data format is incompatible with ours, preventing direct fine-tuning. In this study, we chose to evaluate RadFM and M3D based on their published results in the literature. Future work may involve fine-tuning these models on our dataset for a more direct comparison.

### B.3.3    Impact of Temperature on Performance

For the Impact of Temperature on Performance phase, we employed the 3D-CT-GPT++ (SFT+DPO) model form described in Appendix B.3.1. This version was selected as the final configuration of the 3D-CT-GPT++ model. We conducted experiments varying the temperature parameter from 0.4 to 0.9 to assess its effect on the model's report generation capabilities.

### B.3.4    Impact of Data Quantity on DPO Performance

Both 3D-CT-GPT++ (1) and 3D-CT-GPT++ (SFT) followed the training setup outlined in Appendix B.3.1. The pre-training and fine-tuning phases for 3D-CT-GPT++ (2) were identical to 3D-CT-GPT++ (1). The key distinction lies in the DPO phase, where a refined data selection strategy (Strategy 2) was employed. In Strategy 2, the aim was to enhance contrast between positive and negative examples by selecting the highest-scoring candidate as the positive example and the lowest-scoring candidate as the negative example. If no candidate scored $\geq 3$, the real report was used as the positive example. This strategy was designed to ensure that the selected examples better represented model preferences and human-aligned outputs, leading to more effective optimization during the DPO phase.DPO training for 3D-CT-GPT++ (2) was conducted under the same experimental configuration and batch settings as 3D-CT-GPT++ (1), ensuring consistency across all experiments. For details regarding Strategy 1, please refer to section 3.3.1, which provides an in-depth explanation of how GPT-4 was used for scoring and constructing the preference dataset.

### B.4    Ablation Study Experimental Setup

In the ablation study conducted during the Direct Preference Optimization (DPO) phase, we adhered to the experimental framework detailed in Appendix B.3.1. The primary objective of this study was to evaluate the impact of various model components, with the specific outcomes presented in Table 6. The experimental configurations are described below:

- **Baseline Configuration (3D-CT-GPT++, CTViT)**: The original 3D-CT-GPT++ model, utilizing the CTViT encoder, was trained on the Dataset-XY (train) following the procedure outlined in Section B.3.2.

- **3D-CT-GPT++ with CT-RATE and LoRA (Configuration (b))**: The encoder (CTViT-V) was first trained on the public CT-RATE (train) dataset for 4 batches over 10,000 steps, which required 13 hours. Subsequently, pre-training was performed on CT-RATE (train) for 5 epochs with a single batch, taking 3.8 hours. Finally, LoRA fine-tuning was applied to Dataset-XY (train), lasting 1.4 hours.

- **LoRA Fine-tuning Only (Configuration (e))**: This setup involved applying the LoRA-based fine-tuning method directly to the 3D-CT-GPT++ model. The model was fine-tuned for 2 epochs on Dataset-XY (train) over approximately 1.4 hours, followed by evaluation on Dataset-XY (test) as described in Appendix B.3.1.

- **Refined DPO Model (3D-CT-GPT++ DPO (2))**: The refined DPO model was trained using the same setup as 3D-CT-GPT++ DPO (1), but with a more selective dataset for DPO training based on Strategy 2 outlined in Section B.3.4.

- **Ablation Configuration (f) - Unfreezing the Multi-Layer Perceptron (MLP)**: In this configuration, the MLP was unfrozen during the DPO phase. The model training followed the experimental setup in Appendix B.3.1, utilizing a dataset generated via Strategy 2 for the DPO phase.

- **Ablation Configuration (g) - Adjusted Learning Rate ($3 \times 10^{-7}$)**: Here, the learning rate was set to $3 \times 10^{-7}$ during the DPO phase. The dataset used for DPO training was the same as that in Strategy 2.

Each experimental configuration was replicated five times to ensure the robustness and reliability of the results. All experiments maintained consistent batch sizes and learning rate settings, employing the Adam optimizer throughout the training process. The averaged results from these repetitions are presented in Table 6, highlighting the performance variations attributable to each configuration.

# C  ABLATION STUDY

The ablation study for 3D-CT-GPT++ provides valuable insights into the contributions of different components in the model architecture, training strategies, and hyperparameter settings. Below, we present a detailed analysis of the key findings from our experiments, as shown in Table 6.The comprehensive experimental configuration process is detailed in Section B.4.

## C.1  ENCODER ARCHITECTURE IMPROVEMENTS

First, we compare the improvements in the encoder architecture. The performance difference between models (a) and (e) reflects the impact of the original CTViT encoder versus our enhanced version, CTViT-V. While model (a) used the original CTViT encoder, model (e) employed the improved CTViT-V. The results demonstrate that CTViT-V achieved significant enhancements in metrics such as BLEU-1, BLEU-4, and METEOR. Specifically, BLEU-1 reached 55.98, reflecting a 7.3% increase compared to model (a)'s 52.17. These findings indicate that our improved CTViT-V encoder effectively enhances the model's ability to represent 3D imaging information.

### C.1.1  COMPARISON OF ENCODER ARCHITECTURES IN TERMS OF COMPUTATIONAL RESOURCES AND PERFORMANCE

Table 5: Comparison of Encoder Architectures: 3DViT, CTViT, and CTViT-V in terms of computational resources and performance metrics. All models are based on the CT-CLIP architecture and trained on the same dataset. Best results are in **bold**, and second-best results are underlined.

| Encoder | Batch Size | Time Consumed | Steps | Memory Usage (GB) |
|---------|-----------|---------------|-------|-------------------|
| **3DViT** | 2 | - | 16k | 36 |
| **CTViT** | 8 | 31 hours | 16k | 31.3 |
| **CTViT-V** | 8 | 33 hours | 16k | 30.6 |

To evaluate the trade-offs among different encoder architectures, we compared 3DViT, CTViT, and the proposed CTViT-V in terms of computational efficiency and performance metrics. All models are based on the CT-CLIP framework and trained on an identical proprietary dataset, ensuring a rigorous and fair comparison. As shown in Table 5, CTViT-V achieves a competitive balance between computational resource usage and performance. It maintains efficiency by utilizing a batch size of 8 with slightly longer training time (33 hours) but reduced memory consumption (30.6 GB). These findings highlight the utility of CTViT-V as a robust and scalable encoder, particularly for large-scale 3D medical imaging tasks where both performance and resource optimization are critical. By striking an effective balance between these factors, CTViT-V demonstrates its potential for practical deployment in high-demand settings.

## C.2  DATASET SELECTION

Next, we examine the impact of dataset selection by comparing models (b) and (e). Model (b) was pretrained on the publicly available CT-RATE dataset and fine-tuned on the proprietary dataset, whereas model (e) was trained entirely on the proprietary dataset. The results indicate that model (e), using only the proprietary dataset, achieved better performance across all evaluation metrics, particularly in BLEU-1, where model (e) scored 61.43 compared to model (b)'s 55.98. This suggests that training exclusively on the proprietary dataset captures more effective features and details, significantly enhancing the model's performance.

## C.3  LANGUAGE MODEL SELECTION

The comparison between models (c) and (e) demonstrates the impact of different language models on downstream tasks. Model (c) employed LLaMA2, while model (e) used the Vicuna language model. Although LLaMA2 achieved a higher BLEU-4 score of 13.06 compared to model (e)'s 10.50, model (e) obtained higher scores in other metrics. This suggests that Vicuna is more suitable for medical

Table 6: Ablation study results for 3D-CT-GPT++, highlighting the effects of encoder architecture, dataset choice, language models, MLP freezing, learning rate adjustments, and fine-tuning strategies (SFT+DPO vs. LoRA+DPO). Best results are in **bold**, second-best are underlined.

| Ablation Study | BLEU-1 | BLEU-4 | ROUGE-1 | ROUGE-2 | ROUGE-L | METEOR |
|---|---|---|---|---|---|---|
| **LoRA Phase** | | | | | | |
| **(a)** 3D-CT-GPT++ (CTViT + LoRA) | 52.17 | 11.49 | 0.4711 | 0.2224 | 0.3353 | 0.3308 |
| **(b)** 3D-CT-GPT++ (CT-RATE + LoRA) | **61.43** | 4.58 | 0.3364 | 0.1643 | 0.2599 | 0.1791 |
| **(c)** 3D-CT-GPT++ (LLaMA2 + LoRA) | 45.88 | **13.06** | **0.4882** | **0.2277** | **0.3393** | **0.3684** |
| **(e)** 3D-CT-GPT++ (LoRA) | 55.98 | 10.50 | 0.4561 | 0.2209 | 0.3306 | 0.3061 |
| **DPO Phase** | | | | | | |
| **(f)** Unfreeze MLP**(2)** | 54.89 | 12.46 | 0.4847 | 0.2336 | 0.3496 | 0.3405 |
| **(g)** Learning Rate $3 \times 10^{-7}$**(2)** | 55.76 | 10.81 | 0.4644 | 0.2197 | 0.3310 | 0.3134 |
| **3D-CT-GPT++DPO(2)** | 55.38 | 12.41 | 0.4857 | 0.2309 | 0.3479 | 0.3339 |
| **3D-CT-GPT++DPO(1)** | **56.76** | **13.32** | **0.5117** | **0.2467** | **0.3692** | **0.3542** |
| **(h)** 3D-CT-GPT++ (LoRA+DPO)**(1)** | 54.85 | 12.54 | 0.4881 | 0.2364 | 0.3515 | 0.3384 |

imaging text generation tasks, offering more stable overall performance despite the lower BLEU-4 score.

## C.4 TRAINING STRATEGIES

### C.4.1 FINE-TUNING METHODS: SFT+DPO VS. LORA+DPO

Regarding model training strategies, we analyzed the impact of SFT+DPO versus LoRA+DPO. SFT represents full-parameter fine-tuning, while LoRA adjusts parameters using low-rank adaptation. The comparison between model (h) and 3D-CT-GPT++ DPO(1) shows that SFT+DPO outperformed LoRA+DPO across metrics, particularly with a 3.1% improvement in BLEU-1. This indicates the stronger adaptability of full-parameter fine-tuning in capturing task-specific features. However, LoRA+DPO remains a practical alternative due to its efficiency and comparable performance.

### C.4.2 TRANSITION TO THE DPO PHASE: MLP UNFREEZING AND LEARNING RATE ADJUSTMENT

Additionally, during the transition to the DPO phase, we studied the impact of unfreezing the MLP and adjusting the learning rate on model performance. The comparison between model (f) and 3D-CT-GPT++ DPO(2) shows that unfreezing the MLP led to significant improvements in BLEU-4 and ROUGE metrics, particularly ROUGE-L, which increased from 0.3479 to 0.3496. Conversely, adjusting the learning rate in model (g) resulted in a decrease in performance, indicating that the original learning rate setting was more appropriate. Overall, these comparisons validate that gradually unfreezing the MLP and maintaining appropriate learning rate settings are key factors in enhancing DPO training effectiveness.

## C.5 SUMMARY OF ABLATION RESULTS

These experimental results comprehensively demonstrate the impact of different design choices on 3D-CT-GPT++ in the task of generating 3D medical imaging reports. Through an in-depth analysis of the encoder, dataset, language model, and training strategies, we have identified key design choices that significantly improve generation quality, particularly in quantitative metrics such as BLEU, ROUGE, and METEOR.

# D  QUALITATIVE ANALYSIS

## QUALITATIVE ANALYSIS

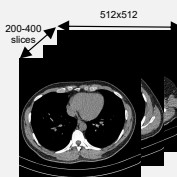

**Prompt:**
Please generate a medical report for this CT scan.

**Real Report:**
The bronchovascular bundles of both lungs are clear, and a few strips of increased density are seen in the middle lobe of the right lung and the lingual segment of the left lung. The bronchus of each lobe and segment are unobstructed. There are no enlarged lymph nodes in the mediastinum and hilus of the lungs. There is no obvious abnormality in the heart and major blood vessels. There is no thickening of the bilateral pleura, and no effusion in the bilateral thoracic cavity. Multiple low-density shadows were seen in the liver, and point-like high-density shadows were seen in the right lobe of the liver.

**3D-CT-GPT:**
The bronchovascular bundles of both lungs became thicker, the transparency of the lung tissue increased, and the bronchus were unobstructed in each lobe and segment. There were no enlarged lymph nodes in the mediastinum and hilus of lions, some calcified lesions in their aorta and coronary arteries, no obvious abnormalities in other heart and major blood vessels, local thickening of bilateral pleura, with no effusion in bilate thoracic cavity.

**3D-CT-GPT++(LORA):**
The bronchovascular bundles of both lungs are clear, no obvious major lesions are seen, the bronchus of each lobe and segment are unobstructed, multiple small to slightly large lymph nodes can be seen in the mediastinum, dense shadows can seen on the walls of the aorta and coronary arteries, and no thickening of bilateral pleura, pleural effusion on both sides, or pleuritic traction on bilatersal sides.

**3D-CT-GPT++(SFT):**
The bronchovascular bundles of both lungs are clear, and nodules are scattered in both upper linders. The largest one is a solid nodule located in the outer basal segment of the lower lobe of right lung. A few string-like increased density shadows can be seen in some areas of lower lung. No enlarged lymph nodes are found in mediastinum and hilus of lunch. Some aorta and coronary arteries are calcified. There is no obvious effusion in bilateral thoracic cavity.

**3D-CT-GPT++(DPO):**
The bronchovascular bundles of both lungs are clear, no obvious major lesions are seen, the bronchus of each lobe and segment are unobstructed, a few cord shadows are visible in the middle lobes of the right lung and left lung, and no enlarged lymph nodes are found in mediastinum and hilus of bilateral lung. No obvious abnormalities are observed in heart and major blood vessels. There is no thickening of pleura and effusion in thoracic cavity.

Figure 5: Comparison of generated reports from different versions of 3D-CT-GPT++ (DPO, SFT, LoRA, and baseline) with the real medical report. Correct answers and hallucinations are highlighted. Each model generated a report based on the same CT scan, highlighting differences in clinical accuracy, detail, and medical terminology. The real report serves as a reference for assessing alignment with actual clinical observations.

