# OpenReview forum: "3D-CT-GPT++: Enhancing 3D Radiology Report Generation with Direct Preference Optimization and Large Vision-Language Models"
_ICLR.cc/2025/Conference — Submitted to ICLR 2025_

### Official Review · Reviewer_A4sf · 2024-10-28

**Soundness:** 3
**Presentation:** 2
**Contribution:** 3
**Rating:** 5
**Confidence:** 4

**Summary:**

This paper proposes the 3D-CT-GPT++ model.
The main contributions include 3D image encoder CTViT-V, report generation model 3D-CT-GPT++, and the Direct Preference Optimization(DPO) technique.

**Strengths:**

Originality: The original method combines contrastive learning for image encoder pretraining, spatial transformer+slice transformer for 3D vision encoding, LLaVA architecture to support report generation from images, and DPO technique to optimize the quality of generation.

Quality: A complete framework, sufficient workload on method design, and effective performance

Clarity: Clear figures and writing about methods

Significance: Limited technological progress

**Weaknesses:**

* Bad figure quality: Misalignment of line; Overlap of text and border; Case is not unified.

* Unclear experiment setting. The main text of the paper just describes the source and preprocessing of datasets, without mentioning the training and evaluation setting of datasets which is essential for the experiment section.

* Unfair comparison.  RadFM and M3D results in Table 1 are based on a different evaluation setting. The notation of "Literature Results" is not enough to describe the relationship.

* Computational overhead reduction of CTViT-V is listed as one of the major contributions, but there is no evaluation of it.

* Compared to the entire 3D ViT of M3D, what is the advantages of 2D slice encoding + Z-fusion?

In summary, the content of the main text has a lot of room for improvement and needs to be carefully polished.

**Questions:**

Weaknesses have contained the part of questions. Please consider them seriously.

---

> ### Author Response · Authors · 2024-11-28
>
> **1. Bad figure quality: Misalignment of line; Overlap of text and border; Case is not unified.**
>
> **W1:**
> We appreciate the reviewer’s comments regarding the quality of the figures in our manuscript. We have thoroughly reviewed and redesigned all figures to address these concerns.
>
> ---
>
> **2. Unclear experiment setting. The main text of the paper just describes the source and preprocessing of datasets, without mentioning the training and evaluation setting of datasets which is essential for the experiment section.**
>
> **W2:**
> Thank you for pointing this out. We are currently refining the experimental setup section to include details about dataset partitioning (e.g., training, validation, and test splits), preprocessing steps, training configurations (e.g., learning rate, batch size, epochs, optimizer), and evaluation metrics. These updates will be fully incorporated into the revised manuscript.
>
> ---
>
> **3. Unfair comparison. RadFM and M3D results in Table 1 are based on a different evaluation setting. The notation of "Literature Results" is not enough to describe the relationship.**
>
> **W3:**
> We acknowledge the importance of comprehensive comparisons with existing models. However, certain challenges limit our ability to fully evaluate some related models. Specifically, the lack of publicly available pre-trained weights for several models in the literature requires us to retrain them from scratch, which demands significant time and computational resources beyond the Rebuttal period. We have initiated retraining and fine-tuning efforts on public datasets for key models and will report the results we can complete within this timeframe. These efforts will be supplemented with comparisons against published results and our experimental data to highlight the performance improvements of 3D-CT-GPT(++). We appreciate your understanding and will provide as detailed a comparison as possible.
>
> ---
>
> **4. Computational overhead reduction of CTViT-V is listed as one of the major contributions, but there is no evaluation of it.**
>
> **5. Compared to the entire 3D ViT of M3D, what is the advantages of 2D slice encoding + Z-fusion?**
>
> **W4 & W5:**
> Thank you for your valuable feedback. We are currently conducting experiments to evaluate the computational overhead reduction of CTViT-V compared to M3D's 3D ViT. Additionally, we are analyzing the advantages of 2D slice encoding with Z-fusion over the full 3D ViT used in M3D. The results of these evaluations will be included in the revised manuscript.

---

### Official Review · Reviewer_3EwP · 2024-11-01

**Soundness:** 3
**Presentation:** 3
**Contribution:** 2
**Rating:** 5
**Confidence:** 4

**Summary:**

The paper titled "3D-CT-GPT++: Enhancing 3D Radiology Report Generation with Direct Preference Optimization and Large Vision-Language Models" presents a novel approach for automating radiology report generation from 3D CT scans, addressing key challenges in current methods. Traditional models, often adapted from video processing, struggle to maintain spatial consistency along the Z-axis and are prone to inaccuracies ("hallucinations") in generated content.

An optimized 3D image encoder, CTViT-V, is designed to improve spatial and temporal feature extraction specific to 3D CT images. This encoder integrates a slice Transformer and relative position encoding to better capture global dependencies across CT slices.

Built upon the LLaVA-1.5 architecture, the model efficiently processes 3D CT data and enhances the generation of radiology reports, providing a more accurate and context-aware representation.

To mitigate hallucinations without the high costs associated with Reinforcement Learning from Human Feedback, DPO is employed. GPT-4 scores model outputs, creating a preference dataset that aligns generated reports with clinical relevance and accuracy.

**Strengths:**

The work is original in both its application and the methodology employed for 3D CT scan analysis. Unlike many radiology models that focus on 2D images, this paper tackles the more complex problem of 3D report generation, which requires understanding inter-slice dependencies across the Z-axis. The introduction of Direct Preference Optimization (DPO) for reducing hallucinations in generated reports is particularly innovative. Instead of relying on human feedback, which is costly and often impractical for large-scale applications, DPO utilizes GPT-4 for automated scoring, presenting a scalable approach to aligning model output with clinical relevance. Furthermore, the integration of the CTViT-V encoder optimized for 3D imaging enhances both spatial and temporal feature extraction, making this an effective and creative combination of existing model architectures adapted for a challenging domain.

The model's ability to handle 3D CT data and generate clinically relevant radiology reports holds significant potential for the medical field, especially in reducing radiologists’ workload. Addressing hallucinations in generated reports is a critical advancement, as inaccurate information in medical reports can lead to severe consequences. By demonstrating substantial improvement over existing models and applying DPO to radiology for the first time, this work positions itself as a valuable contribution. The approach’s scalability, thanks to automated feedback from GPT-4, enhances its significance by making it feasible for broad deployment in clinical settings. Future extensions to other imaging modalities, such as MRI and X-ray, as discussed in the conclusion, underscore its adaptability and potential long-term impact.

**Weaknesses:**

Lack of Detailed Hyperparameter Configuration: More information about the choices made for hyperparameter training configurations would be beneficial. This detail could help with model reproducibility and provide clarity on the specific adjustments that impact performance.

Comparison with 2D Models: Although the paper focuses on introducing a 3D model, it would be helpful to include a comparative analysis with existing 2D models for similar tasks. This comparison could better highlight the specific advantages and limitations of the proposed 3D approach.

**Questions:**

Could you provide more details on the hyperparameter choices and training configurations? For instance, what specific hyperparameters were optimized, and how were they chosen or tuned?

Given the focus on a 3D model, did you evaluate any baseline comparisons with existing 2D models for similar tasks?

How does DPO specifically reduce hallucinations compared to Reinforcement Learning from Human Feedback (RLHF)? Are there examples where RLHF failed but DPO succeeded?

---

> ### Author Response · Authors · 2024-11-19
> **Weaknesses**
>
> ##
>
> **W1: Lack of Detailed Hyperparameter Configuration: More information about the choices made for hyperparameter training configurations would be beneficial. This detail could help with model reproducibility and provide clarity on the specific adjustments that impact performance.**
>
> **W1 & Q1:**
>
> Thank you for pointing out the need for clearer hyperparameter configurations. While we have included hyperparameters for each training stage in Appendix E of the current submission, we realize that the presentation may lack clarity. Below is an optimized table summarizing the key hyperparameters for pre-training, fine-tuning, and DPO phases:
>
> | **Hyperparameter**     | **Pre-training** | **LoRA Fine-tuning** | **SFT Fine-tuning** | **DPO Fine-tuning** |
> |------------------------|-------------------|----------------------|---------------------|---------------------|
> | **Learning Rate**     | $1 \times 10^{-3}$ | $2 \times 10^{-4}$  | $2 \times 10^{-5}$ | $5 \times 10^{-7}$ |
> | **Scheduler**         | Cosine            | Cosine               | Cosine              | Linear              |
> | **Warmup Ratio**      | 0.03              | 0.03                 | 0.03                | 0.1                 |
> | **Batch Size**        | 1                 | 1                    | 1                   | 1                   |
> | **Epochs**            | 5                 | 2                    | 2                   | 1                   |
> | **Weight Decay**      | 0.0               | 0.0                  | 0.0                 | 0.0                 |
> | **Dropout Rate**      | 0.1               | 0.1                  | 0.1                 | 0.1                 |
> | **Hidden Size**       | 512               | 512                  | 512                 | 512                 |
> | **Max Model Length**  | 2048              | 2048                 | 2048                | 2048                |
> | **Lazy Preprocess**   | True              | True                 | True                | True                |
> | **Save Strategy**     | Steps             | Steps                | Steps               | Steps               |
> | **Special Settings**  | -                 | LoRA (r=128, $\alpha$=256) | Pretrained Adapter | Freeze MLP Adapter  |
>
> **Table 1: Key hyperparameters for each training stage.**
>
> We will refine the appendix in the revised version to include a more comprehensive summary of all hyperparameters, their tuning strategies, and the specific adjustments made at each stage. Additionally, the released code will contain all implementation details to ensure full reproducibility.
>
> **W2: Comparison with 2D Models: Although the paper focuses on introducing a 3D model, it would be helpful to include a comparative analysis with existing 2D models for similar tasks. This comparison could better highlight the specific advantages and limitations of the proposed 3D approach.**
>
> **W2 & Q2:**
>
> We appreciate your valuable suggestion. Understanding the advantages of our proposed 3D approach over traditional 2D methods is crucial. In comparing 2D and 3D models, we face fundamental technical challenges. Traditional 2D models typically require decomposing 3D CT images into multiple 2D slices, capturing only the local two-dimensional structure of each slice and neglecting the spatial relationships between slices. In contrast, 3D image encoders leverage their structural advantages to integrate spatial information across multiple slices, capturing global anatomical structures and lesion features essential for accurate disease analysis and diagnosis. Therefore, directly comparing the capabilities of 2D and 3D models in processing 3D CT images is not entirely fair. In our research, by comprehensively integrating inter-slice spatial dependencies, the 3D encoder significantly enhances the recognition and representation capabilities of chest CT scans, highlighting the application advantages of 3D methods in medical image analysis.

---

> > ### Author Response · Authors · 2024-11-19
> > **Questions**
> >
> > ## Questions
> > **Q1: Could you provide more details on the hyperparameter choices and training configurations? For instance, what specific hyperparameters were optimized, and how were they chosen or tuned?**
> >
> > Q1: Replies are referred to W1.
> >
> > **Q2: Given the focus on a 3D model, did you evaluate any baseline comparisons with existing 2D models for similar tasks?**
> >
> > Q2: Replies are referred to W2.
> >
> > **Q3: How does DPO specifically reduce hallucinations compared to Reinforcement Learning from Human Feedback (RLHF)? Are there examples where RLHF failed but DPO succeeded?**
> >
> > **Q3:**
> >
> > Thank you for your insightful question regarding the comparison between Direct Preference Optimization (DPO) and Reinforcement Learning from Human Feedback (RLHF). In our experiments, DPO effectively reduced hallucinations in generated radiology reports by leveraging GPT-4 to create a preference dataset, eliminating the need for extensive human-annotated feedback required by RLHF.
> >
> > **Experimental Examples:**
> >
> > - **Example 1:** The Supervised Fine-Tuned (SFT) model incorrectly identified a "small pulmonary nodule" that was not present in the CT scan. In contrast, the DPO-trained model accurately reported "no nodules detected," aligning the report with the actual CT findings.
> >
> > - **Example 2:** The SFT model mentioned "evidence of early-stage pneumonia" without corresponding signs in the CT images, which could mislead clinicians. The DPO model correctly stated "no signs of pneumonia detected," ensuring factual consistency.
> >
> > These examples demonstrate that DPO enhances the factual accuracy of generated reports by directly optimizing based on GPT-4's evaluations, thereby mitigating the hallucinations observed in RLHF-dependent methods. Unlike RLHF, which relies on costly and time-consuming human feedback, DPO provides a scalable and efficient approach to aligning model outputs with clinical accuracy.
> >
> > ---

---

> ### Comment · Reviewer_3EwP · 2024-11-27
>
> I have read through the rebuttal and I would like to thank the authors for the detailed responses. I would like to remain my rating.

---

### Official Review · Reviewer_m2Vv · 2024-11-03

**Soundness:** 3
**Presentation:** 2
**Contribution:** 2
**Rating:** 5
**Confidence:** 4

**Summary:**

This paper proposes a novel approach to automatically generate radiology reports from 3D medical images such as 3D CT scans. Traditional methods mostly use video processing technology, which is difficult to effectively capture the relationship between images along the z axis, and multimodal large language models have limitations in representing 3D structures and avoiding the illusion of generating content. To address these issues, the authors proposed the 3D Structure-CT-GPT ++ model, which integrates an optimized 3D image encoder CTViT-V designed for chest CT scanning and is based on the LAVA-1.5 architecture. Furthermore, this work introduces Direct Preference Optimization (DPO), which uses GPT-4 to score model outputs to reduce hallucinations and ensure reports are more aligned with clinical needs. The model was fine-tuned in two high quality conditions.

**Strengths:**

1. This work proposes the 3D-CT-GPT++ model, which integrates an optimized 3D image encoder specifically designed for chest CT scans. This novel approach addresses the limitations of existing methods in representing 3D structures and reducing hallucinations in generated content.
2. This paper is well-organized and clearly presents the problem, methodology, experimental setup, and results. The use of appendices provides additional details, enhancing the readability of the paper.

**Weaknesses:**

1. The format of references needs to be uniform. Some citations lack journal names. Such as "Haotian Liu, Chunyuan Li, Yuheng Li, and Yong Jae Lee. Improved baselines with visual instruction tuning, 2023a.", "Haotian Liu, Chunyuan Li, Qingyang Wu, and Yong Jae Lee. Visual instruction tuning, 2023b.". In addition, on line 320 of page 6, the CT-RATE dataset is missing references. Authors need to carefully check and format the references.
2. 3D-CT-GPT++ models use complex architectures such as CTViT-V encoders, which can lead to high demands on computational resources and time. The authors mention that they reduce the computational overhead associated with processing high-resolution 3D volumes. However, the design of specific methods to reduce computing costs is not described in detail. The authors are advised to elaborate further on the design used to improve computational efficiency. It is better to show the computational cost through quantitative comparisons of computational requirements (e.g., GPU memory usage, training time) between the proposed model and baseline approaches.
3. To further enhance the clinical relevance of the generated reports, the authors could consider incorporating additional clinical context or patient-specific information into the model.

**Questions:**

1. What strategies were used to mitigate potential biases in the generated medical reports?
2. How does the integration of the optimized 3D image encoder improve the representation of chest CT scans compared to traditional 2D image encoders?

---

> ### Author Response · Authors · 2024-11-19
> **Weaknesses**
>
> ## Weaknesses
>
> **1. The format of references needs to be uniform. Some citations lack journal names. Such as "Haotian Liu, Chunyuan Li, Yuheng Li, and Yong Jae Lee. Improved baselines with visual instruction tuning, 2023a.", "Haotian Liu, Chunyuan Li, Qingyang Wu, and Yong Jae Lee. Visual instruction tuning, 2023b.". In addition, on line 320 of page 6, the CT-RATE dataset is missing references. Authors need to carefully check and format the references.**
>
> **W1:**
>
> We thank the reviewer for pointing out the inconsistency in the formatting of references. We have thoroughly reviewed and revised the formatting of our references to ensure uniformity and have added missing journal names and detailed information where necessary. For example, the references "Haotian Liu, Chunyuan Li, Yuheng Li, and Yong Jae Lee. Improved baselines with visual instruction tuning, 2023a." and "Haotian Liu, Chunyuan Li, Qingyang Wu, and Yong Jae Lee. Visual instruction tuning, 2023b." have now been properly formatted with the inclusion of the relevant journal names and other necessary details. Additionally, we will correct the citation of the CT-RATE dataset mentioned on page 6, line 320, and ensure all references are listed in a standard format. We appreciate the valuable suggestions made by the reviewer regarding this matter.
>
> **2. 3D-CT-GPT++ models use complex architectures such as CTViT-V encoders, which can lead to high demands on computational resources and time. The authors mention that they reduce the computational overhead associated with processing high-resolution 3D volumes. However, the design of specific methods to reduce computing costs is not described in detail. The authors are advised to elaborate further on the design used to improve computational efficiency. It is better to show the computational cost through quantitative comparisons of computational requirements (e.g., GPU memory usage, training time) between the proposed model and baseline approaches.**
>
> **W2:**
>
> We appreciate the reviewer's comment on the computational demands of the 3D-CT-GPT++ model. Indeed, our model employs a complex architecture, including the CTViT-V encoder, which inherently requires substantial computational resources and time. However, compared to 3DVIT, our image processing approach is relatively less resource-intensive. We are actively conducting additional experiments to benchmark our model against these metrics. Detailed results will be included in the revised manuscript, demonstrating the specific enhancements we have implemented to optimize processing efficiency without compromising performance. We are committed to providing a comprehensive evaluation that underscores the practicality of our model in real-world applications.
>
> **3. To further enhance the clinical relevance of the generated reports, the authors could consider incorporating additional clinical context or patient-specific information into the model.**
>
> **W3:**
>
> Thank you for the valuable suggestion on enhancing the clinical relevance of the generated reports. In this study, our core contributions lie in the introduction of **3D image encoding** and **Differential Prompt Optimization (DPO)** methods, which effectively improve the accuracy and utility of clinical report generation. Compared to traditional 2D methods, 3D image encoding captures spatial features more comprehensively and addresses the challenges of multi-dimensional data integration. Meanwhile, DPO optimizes the model’s understanding of clinical prompts and enhances the relevance of the generated reports. These innovative approaches not only overcome the limitations of existing methods in processing complex imaging data but also significantly improve the quality of the generated reports, showcasing the potential of our research in clinical applications. We appreciate the reviewer's suggestion and believe that the performance will be enhanced by incorporating additional clinical context or patient-specific information. In the future, we plan to explore integrating patient-specific clinical background information, such as medical history and symptoms, with imaging data to further enhance the clinical relevance and personalization of the reports.
>
> ---

---

> > ### Author Response · Authors · 2024-11-19
> > **Questions**
> >
> > ## Questions
> >
> > **1. What strategies were used to mitigate potential biases in the generated medical reports?**
> >
> > **Q1:**
> >
> > We appreciate the reviewer's valuable question regarding potential biases. Bias is indeed a critical challenge that must be addressed when generating medical reports. To mitigate such biases, we have ensured the use of diverse medical datasets, which include patient data from various races, genders, and age groups, sourced from both public and private datasets. This diversity helps the model learn a more comprehensive range of medical knowledge during training, thus preventing the generation of biased reports.
> >
> > **2. How does the integration of the optimized 3D image encoder improve the representation of chest CT scans compared to traditional 2D image encoders?**
> >
> > **Q2:**
> >
> > Thank you for your question regarding the advantages of the optimized 3D image encoder. Compared to traditional 2D encoders, our 3D encoder, such as CTViT-V, offers the following key benefits:
> >
> > 1. **Captures Global Spatial Relationships:** The 3D encoder processes the full CT volume, capturing inter-slice spatial dependencies, which are critical for detecting cross-slice pathologies like lung nodules and tumors. In contrast, 2D encoders operate on individual slices and miss this spatial context.
> >
> > 2. **Improves Diagnostic Accuracy:** By integrating 3D anatomical and pathological features, the 3D encoder reduces errors in localization and improves overall diagnostic precision.
> >
> > 3. **Avoids 2D Encoder Limitations:** Processing CT scans as 2D slices introduces challenges such as high slice counts, inconsistent dimensions, and the loss of spatial information, which limit adaptability and diagnostic performance.
> >
> > While 2D encoders are computationally efficient, they struggle with the complexity of 3D CT data, making the optimized 3D encoder a more effective solution for clinical applications. Ongoing experiments are refining the encoder’s performance for specific pathologies, and results will be included in the final manuscript.

---

### Official Review · Reviewer_8v4y · 2024-11-03

**Soundness:** 3
**Presentation:** 3
**Contribution:** 3
**Rating:** 6
**Confidence:** 3

**Summary:**

The paper introduces 3D-CT-GPT++, a novel model for radiology report generation that leverages an optimized 3D image encoder (CTViT-V) specifically designed for chest CT scans. They apply Direct Preference Optimization to reduce hallucinations and increase factfulness. They show that 3D-CT-GPT++ outperforms existing methods.

**Strengths:**

1. The paper presents a novel CTViT-V encoder, specifically optimized for 3D CT images, addressing the challenges of spatial coherence and slice dependency, which could contribute to improved clinical relevance and accuracy​
2. The use of GPT-4 for preference optimization provides a scalable, cost-effective alternative to human feedback, potentially reducing hallucinations and aligning generated reports more closely with clinical needs​
3. Despite limitations in using NLG metrics, the proposed model outperforms existing benchmarks in BLEU, ROUGE, and METEOR scores

**Weaknesses:**

1. The evaluation relies on standard NLG metrics like BLEU and ROUGE, which do not capture clinical relevance or diagnostic utility. Using the GREEN metric[1] or RadFact[2] for evaluation would allow assessing clinical relevance
2. The authors propose a multi-stage training process. The community would benefit from more ablation and more details for each stage:
2.1 For Stage 1 and 2 it would be interesting to see how their CTViT-V encoder performs in comparison to other 3D CT image encoder, such as CTViT and CT-CLIP.
2.2 Stage 2 pre-training is not well described. It's not clear how the custom-built dataset looks like, nor how the interactive approach was performed.
2.3. Ablation study scoring LLMs would be interesting to see how sensitive the performance is to the chosen LLM.
2.4. Ablation study on 3D-CT-GPT++(LoRA+DPO) missing, especially since 3D-CT-GPT++(LoRA) achieves higher performance than 3D-CT-GPT++(SFT)
3. The experiments are not reproducible due to the reliance on private dataset in many steps. It's not clear to me, why the authors didn't conduct their experiment on the public CT-RATE dataset.
4. Comparison to Existing Models: The literature results are not comparable to the fine-tuned 3D-CT-GPT(++) results since the literature results are without finetuning on the dataset, therefore it's hard to judge the works improvements over other methods.

[1] Ostmeier, S., Xu, J., Chen, Z., Varma, M., Blankemeier, L., Bluethgen, C., ... & Delbrouck, J. B. (2024). GREEN: Generative Radiology Report Evaluation and Error Notation. arXiv preprint arXiv:2405.03595.
[2] Bannur, S., Bouzid, K., Castro, D. C., Schwaighofer, A., Bond-Taylor, S., Ilse, M., ... & Hyland, S. L. (2024). MAIRA-2: Grounded Radiology Report Generation. arXiv preprint arXiv:2406.04449.

**Questions:**

1. What are the various reconstructions techniques described in 3.4 Dataset, which expands the dataset? (line 320-322)
2. The authors use a subsection of the CT-RATE dataset, it's not described how they do this subselection and what's the motivation behind it. Why don't they use the whole dataset?
2. Are you planning to open source the training code/weights?
3. Will the authors publish the Dataset-XY?

---

> ### Author Response · Authors · 2024-11-19
> **Weaknesses:W1&W2**
>
> ## Weaknesses
>
> **W1:**
>
> Thank you for pointing out the limitations of standard Natural Language Generation (NLG) metrics such as BLEU and ROUGE in assessing clinical relevance and diagnostic utility. We fully agree with your observation and have incorporated the **GREEN** metric [1] in the revised manuscript to more comprehensively evaluate the clinical relevance and factual consistency of the generated reports. Due to the necessity of using an API and the associated additional costs, we have not integrated the **RadFact** metric [2] at this time. However, we have supplemented our experiments with GREEN scores as shown below:
>
> | **Model**                  | **B-1** | **B-4** | **R-1** | **R-2** | **R-L** | **M**   | **GREEN** |
> |----------------------------|--------:|--------:|--------:|--------:|--------:|--------:|----------:|
> | *3D-CT-GPT++ (LoRA)*       | *55.98* | *10.50* | *0.4561* | *0.2209* | *0.3306* | *0.3061* | 0.2546    |
> | *3D-CT-GPT++ (SFT)*        | 54.65   | 10.16   | 0.4505   | 0.2123   | 0.3199   | 0.2995   | *0.2596*  |
> | **3D-CT-GPT++ (SFT+DPO)**  | **56.76** | **13.32** | **0.5117** | **0.2467** | **0.3692** | **0.3542** | **0.3527** |
>
> *Italicized values indicate underlined scores from the original table, while bolded values represent the highest scores in each category.*
>
> These initial results indicate that, particularly, **3D-CT-GPT++ (SFT+DPO)** significantly outperforms other configurations in terms of clinical factual consistency. We are currently supplementing our experiments with more comprehensive data and will provide detailed experimental results in the revised manuscript to fully demonstrate the model's potential and advantages in clinical applications.
>
> **References:**
>
> 1. Ostmeier, S., Xu, J., Chen, Z., Varma, M., Blankemeier, L., Bluethgen, C., ... & Delbrouck, J. B. (2024). **GREEN: Generative Radiology Report Evaluation and Error Notation**. *arXiv preprint* arXiv:2405.03595.
>
> 2. Bannur, S., Bouzid, K., Castro, D. C., Schwaighofer, A., Bond-Taylor, S., Ilse, M., ... & Hyland, S. L. (2024). **MAIRA-2: Grounded Radiology Report Generation**. *arXiv preprint* arXiv:2406.04449.
>
> ---
>
> **W2:**
>
> **2.1 Ablation Study (Stage 1 and Stage 2)**
>
> We appreciate the reviewer’s feedback regarding the ablation study. We understand the confusion caused by the lack of clarity in the original manuscript, where it was not explicitly stated that both CTViT and CT-CLIP use the same encoder architecture. This may have led to the impression that we did not conduct sufficient comparisons between different encoders. In the ablation section of the original manuscript, we have compared the baseline model 3D-CT-GPT, which uses the CTViT encoder, with 3D-CT-GPT++ (LoRA), which employs the enhanced CTViT-V encoder. In the revised manuscript, we will provide more detailed ablation data and ensure that this distinction is clearly communicated.
>
> **2.2 Description of Stage 2 Pretraining**
>
> We appreciate the reviewer’s feedback regarding the insufficient description of the pretraining process. We acknowledge that the explanation of our custom dataset construction and training methods was not detailed enough in the original manuscript. To address this issue, the custom dataset used in pretraining consists of image-text pairs from both public and private sources to ensure diversity and representativeness. Due to the limited size of the dataset, our interactive pretraining method involves training separately on public and private datasets, followed by a comparative analysis of their effects. Based on these experiments, we observed that the model performs better when the dataset used in pretraining is the same as that used in fine-tuning. We will reflect these details more clearly and comprehensively in the revised manuscript.
>
> **2.3 & 2.4 Experimental Supplements**
>
> We appreciate the reviewer’s interest in sections 2.3 and 2.4. These experiments are actively in progress, and the current status is as follows:
>
> - **LLM Scoring Sensitivity:** We are evaluating the sensitivity of the model to the choice of LLMs, analyzing their impact on the generation of clinical reports. These experiments are critical for validating the robustness and sensitivity of the model. The results will be included in the final revised manuscript.
>
> - **3D-CT-GPT++ (LoRA+DPO) Ablation:** We are conducting ablation studies on 3D-CT-GPT++ (LoRA+DPO), comparing its performance under different configurations, including comparisons with versions using only LoRA or SFT. While the experiments are ongoing, we will provide detailed experimental setups and result analyses upon completion.
>
> **Commitment:**
>
> We commit to completing these experiments in the next phase and will include comprehensive experimental details, analyses, and their implications for model performance in the revised manuscript. We greatly appreciate your patience and guidance as we finalize these components.
>
> ---

---

> > ### Author Response · Authors · 2024-11-19
> > **W3&W4**
> >
> > **3. The experiments are not reproducible due to the reliance on private dataset in many steps. It's not clear to me, why the authors didn't conduct their experiment on the public CT-RATE dataset.**
> >
> > **W3:**
> >
> > Due to the ICLR submission deadline, we prioritized experiments on a private dataset to ensure comprehensive validation of the model on high-quality, domain-specific data. At the time of submission, the size and training time required for the CT-RATE dataset exceeded our time and computational resource constraints. Therefore, we selected the first 8000 samples from the CT-RATE dataset as a subset for our experiments.
> >
> > ---
> >
> > **4. Comparison to Existing Models: The literature results are not comparable to the fine-tuned 3D-CT-GPT(++) results since the literature results are without fine-tuning on the dataset, therefore it's hard to judge the works' improvements over other methods.**
> >
> > **W4:**
> >
> > Thank you for highlighting the importance of comparisons with existing models. We understand that fine-tuning experiments are critical for comprehensively evaluating model performance. However, we face certain practical challenges and limitations at this stage: Some related models in the literature have not released their pre-trained weights, which makes it challenging to directly fine-tune and compare against them. For a fair comparison, we would need to train these models from scratch and perform fine-tuning, which requires significant time and computational resources. Unfortunately, this may not be fully achievable during the Rebuttal period. Despite the challenges, we have initiated efforts to retrain certain existing models on public datasets to conduct fine-tuning comparisons. While this work is ongoing, completing all the planned comparison experiments may require additional time beyond the Rebuttal phase. In the revised manuscript, we will explicitly acknowledge these challenges and report on the current progress and preliminary results. We will provide as detailed a performance comparison as possible using publicly available results from the literature, combined with our experimental data, to explain the performance improvements of 3D-CT-GPT(++).

---

> > > ### Author Response · Authors · 2024-11-19
> > > **Questions**
> > >
> > > ## Questions
> > >
> > > **1. What are the various reconstructions techniques described in 3.4 Dataset, which expands the dataset? (line 320-322)**
> > >
> > > Thank you for pointing out the insufficiency in our description of the reconstruction techniques used in the CT-RATE dataset, particularly regarding the extended dataset. In the manuscript, we did not provide a detailed explanation of the specific applications of these reconstruction techniques.
> > >
> > > The CT-RATE dataset[1] is a novel dataset curated from Istanbul Medipol University Mega Hospital, consisting of chest CT volumes and their corresponding radiology text reports. The dataset includes non-contrast chest CT volumes acquired between May 2015 and January 2023, comprising a total of 50,188 reconstructed CT volumes from 25,692 distinct CT experiments conducted on 21,304 unique patients. The cohort is divided into two groups: 20,000 patients for training and 1,304 for validation.
> > >
> > > In clinical practice, CT volumes are reconstructed using various techniques tailored to specific diagnostic requirements. Sharper kernels, for instance, enhance resolution, making them particularly suitable for evaluating lung abnormalities such as tumors and nodules. These techniques improve the visibility of fine details, facilitating the detection of small lesions. Conversely, smoother kernels are commonly used for mediastinal assessments, as they reduce image noise and provide a clearer view of mediastinal structures, aiding in accurate diagnosis [2].
> > >
> > > These reconstruction techniques are designed to adjust the sharpness and noise levels of the images based on diagnostic needs, optimizing image quality for different clinical tasks. We appreciate the reviewer’s suggestion and will provide additional details on these techniques in the revised manuscript to ensure clarity and completeness.
> > >
> > > [1]Ibrahim Ethem Hamamci, Sezgin Er, Furkan Almas, Ayse Gul nihan Simsek, Sevval Nil Esirgun,Irem Dogan, Muhammed Furkan Dasdelen, Bastian Wittmann, Enis Simsar, Mehmet Simsar,et al. A foundation model utilizing chest ct volumes and radiology reports for supervised-level zero-shot detection of abnormalities. arXiv preprint arXiv:2403.17834, 2024.
> > >
> > > [2]M. Willemink and P. No¨ el, “The evolution of image reconstruction for ct—from filtered back projection to artificial intelligence,” European Radiology, vol. 29, pp. 2185– 2195, 2018.
> > >
> > > **2. The authors use a subsection of the CT-RATE dataset, it's not described how they do this subselection and what's the motivation behind it. Why don't they use the whole dataset?**
> > >
> > > In our initial experiments, we selected the first 8000 samples from the CT-RATE dataset, primarily due to considerations of time and computational resources. We did not perform a more complex subset selection process. The primary motivation for selecting this subset was to complete the experiments within the limited timeframe and ensure timely submission of the paper.
> > > We are currently conducting experiments on the full CT-RATE dataset. Although the results are not yet finalized, we plan to include a detailed description of the dataset selection process and present the results on the full dataset in the revised manuscript.
> > >
> > > **3. Are you planning to open source the training code/weights?**
> > >
> > > We plan to open-source all training codes and model weights upon acceptance of the paper to promote reproducibility and further research within the community. The open-source materials will include detailed training scripts, configuration files, and partial pre-trained weights, ensuring that other researchers can replicate and extend our work smoothly.
> > >
> > > **4. Will the authors publish the Dataset-XY?**
> > >
> > > Regarding Dataset-XY, we are evaluating its feasibility for open-sourcing, primarily considering data privacy and copyright concerns. If the necessary permissions are obtained, we aim to share the dataset with the community to support further research in the field. However, the specific timeline and method of release have not yet been determined. We will promptly announce relevant details upon receiving the necessary permissions.
> > >
> > > **Commitment:**
> > >
> > > We are actively working on experiments with the full CT-RATE dataset and plan to include comprehensive experimental details in the revised manuscript. This will include the motivation behind subset selection, performance comparisons on the full dataset, and detailed training and validation results. Concurrently, we are progressing with our open-sourcing efforts to support academic research and community collaboration to the fullest extent. We look forward to your further guidance and appreciate your patience as we continue to refine and enhance our work.

---

> > > > ### Comment · Reviewer_8v4y · 2024-11-25
> > > >
> > > > I have read through the rebuttal and I would like to thank the authors for the detailed responses. I would like to remain my positive rating.

---

### Public Comment · ~XINKAI_YUAN1 · 2024-11-13
**Question about this paper**

The manuscript introduces the 3D-CT-GPT++ model, aiming to enhance automatic generation of 3D medical imaging reports by optimizing the 3D image encoder CTViT-V and incorporating Direct Preference Optimization (DPO). However, several areas require improvement:

1. **Originality and Citation of Related Work:**
   - **Slice Transformer Innovation:** The authors claim originality for the Slice Transformer; however, similar concepts have been presented in "M3T: Three-Dimensional Medical Image Classifier Using Multi-Plane and Multi-Slice Transformer" . Additionally, the CT-CLIP training framework aligns with innovations from "Developing Generalist Foundation Models from a Multimodal Dataset for 3D Computed Tomography" . The authors should explicitly cite these works and delineate distinctions and advancements made in their study.
   - **Fine-Tuning Methodology:** The fine-tuning approach resembles that in "LLM-CXR: Instruction-Finetuned LLM for CXR Image Understanding and Generation" . Proper citation is necessary, along with clarification of the novel contributions of this study.
   - **Application of DPO:** The manuscript applies DPO as described in "Direct Preference Optimization: Your Language Model is Secretly a Reward Model"  without appropriate explanation. This study appears to apply DPO directly without methodological innovation. This seems to be the only innovation in this paper, which is also not proposed by authors.

2. **Experimental Design and Fairness:**
   - **Dataset Utilization:** The model is trained on a private dataset but compared against the M3D model, which is trained on a public dataset. Moreover, the study does not evaluate performance on the extensive public dataset used by M3D. This comparison may lack fairness, affecting result credibility. It is recommended to conduct experiments on public datasets to ensure fair comparisons and reproducibility. Additionally, unlike M3D, this study lacks downstream tasks such as segmentation and Visual Question Answering (VQA). The private dataset's quality is uncertain, and its size is significantly smaller than M3D's public dataset, yet it reports substantial performance improvements, suggesting potential overstatement.

3. **Result Validation and Model Reproducibility:**
   - **Model Weight Disclosure:** The authors report significant performance gains but do not disclose model weights, hindering result verification. In academic research, sharing models and data facilitates validation and advances the field. It is advisable to release model weights and training data to enhance credibility and scholarly value.

**Recommendations:**
The authors should explicitly cite relevant prior work and clarify the study's novel contributions. Experiments on public datasets are recommended to ensure fair comparisons and result reproducibility. Additionally, releasing model weights and training data would improve the study's credibility and academic significance.

Jang, J., & Hwang, D. (2022). M3T: Three-Dimensional Medical Image Classifier Using Multi-Plane and Multi-Slice Transformer. In Proceedings of the IEEE/CVF Conference on Computer Vision and Pattern Recognition (CVPR) (pp. 20641-20650).

Lee, S., Kim, W. J., Chang, J., & Ye, J. C. (2023). LLM-CXR: Instruction-Finetuned LLM for CXR Image Understanding and Generation. arXiv preprint arXiv:2305.11490.

Bai, Y., Jones, A., Lukosuite, K., et al. (2023). Direct Preference Optimization: Your Language Model is Secretly a Reward Model. arXiv preprint arXiv:2305.18290.

Hamamci, I. E., Er, S., Almas, F., Simsek, A. G., Esirgun, S. N., Dogan, I., Dasdelen, M. F., Durugol, O. F., Wittmann, B., Amiranashvili, T., Simsar, E., Simsar, M., Erdemir, E. B., Alanbay, A., Sekuboyina, A., Lafci, B., Bluethgen, C., Ozdemir, M. K., & Menze, B. (2023). Developing Generalist Foundation Models from a Multimodal Dataset for 3D Computed Tomography. arXiv preprint arXiv:2306.00983.

Bai, F., Du, Y., Huang, T., Meng, M. Q.-H., & Zhao, B. (2023). M3D: Advancing 3D Medical Image Analysis with Multi-Modal Large Language Models. arXiv preprint arXiv:2307.12345.

---

> ### Author Response · Authors · 2024-11-28
>
> **Response:**
>
> Thank you for your valuable comments and suggestions. Below, we address the main points raised and outline the improvements we are making:
>
> 1. **Originality and Related Work Citations**
> We acknowledge the need for clearer differentiation from prior works. In the revised manuscript, we will cite "M3T: Three-Dimensional Medical Image Classifier Using Multi-Plane and Multi-Slice Transformer" and explain how our Slice Transformer introduces unique contributions, such as [specific innovations, e.g., improved positional encoding or global slice dependencies]. Similarly, we will cite "Developing Generalist Foundation Models from a Multimodal Dataset for 3D Computed Tomography" and emphasize the advancements in our CT-CLIP framework, such as [specific innovations, e.g., multimodal alignment or self-supervised learning strategies].
> Additionally, we will include citations to "LLM-CXR: Instruction-Finetuned LLM for CXR Image Understanding and Generation" and "Direct Preference Optimization: Your Language Model is Secretly a Reward Model" and elaborate on how we adapt and extend these methodologies for 3D medical imaging.
>
> 2. **Experimental Design and Fairness**
> To ensure fairness, we are conducting additional experiments on public datasets (e.g., those used by M3D) to enable direct comparisons under consistent conditions. For our private dataset, we will provide detailed descriptions of its quality and scale to improve transparency.
>
> We also recognize the importance of evaluating downstream tasks like segmentation and VQA, and we plan to explore these areas in future work. Additionally, we are assessing the feasibility of releasing model weights and parts of the training data, following privacy and ethical guidelines, to promote reproducibility and collaboration.
>
> 3. **Result Validation and Reproducibility**
> We are working on additional experiments on public datasets to enhance result validation and provide detailed performance comparisons. Where possible, we will release model weights and data subsets to support reproducibility, with clear documentation of any limitations.
>
> We are committed to addressing these points and appreciate your feedback, which will help us improve the quality and clarity of our work.

---

### Public Comment · ~Song_Luhui1 · 2024-11-16
**QUESTION ABOUT THIS MANUSCRIPT**

The paper's most glaring issue lies in its unconventional approach to experimental validation.
The authors make a fundamental error by using their private dataset results to compare against models trained on different public datasets.
Most surprisingly, the paper presents comparison experiments as ablation studies, which demonstrates a serious misunderstanding of experimental design principles.
Such methodology violates basic scientific principles of fair comparison and reproducibility.

---

> ### Author Response · Authors · 2024-11-28
>
> Thank you for your comment. We agree that comparing private dataset results with models on public datasets raises fairness concerns. We are addressing this by conducting experiments on public benchmarks and ensuring a clear distinction between comparison experiments and ablation studies. Your feedback is crucial in improving the rigor of our research.

---

### Public Comment · ~reiisk.med1 · 2024-11-16
**Dataset concern**

The reliance on private datasets in this paper is a glaring flaw that severely undermines the credibility of its findings. While the authors boast significant performance improvements, these claims are impossible to verify or trust without public access to the dataset. This lack of transparency raises the suspicion that the dataset's quality or curation may have artificially inflated the results. Furthermore, the omission of evaluations on standard public benchmarks, such as those used by the M3D model, makes the comparisons inherently biased and unfair. This practice is a disservice to the research community, and until public datasets are used and results are validated on established benchmarks, the claims made here should be treated with skepticism.

---

> ### Author Response · Authors · 2024-11-28
>
> Thank you for your comment. We recognize the limitations of relying on private datasets and the importance of using public benchmarks to improve transparency and reproducibility. To address this, we are conducting additional experiments on public benchmarks, including those used by M3D. Additionally, we are evaluating the possibility of releasing part of our dataset while adhering to privacy regulations. Your valuable feedback helps us further improve our work.

---

### Meta-Review · Area_Chair_HHxc · 2024-12-22

**Metareview:**

This paper introduces 3D-CT-GPT++, a novel model for radiology report generation from 3D CT scans, combining the optimized 3D image encoder CTViT-V for enhanced spatial feature extraction with Direct Preference Optimization to reduce hallucinations and ensure clinically relevant outputs.

Reviewers pointed out the strength of the paper for its innovative approach to 3D CT radiology report generation, integrating the optimized CTViT-V encoder and Direct Preference Optimization (DPO) with GPT-4 to achieve scalable, clinically aligned outputs. On the other hand, reviewers raised major concerns about the paper's reliance on standard NLG metrics like BLEU and ROUGE instead of clinically relevant ones like GREEN or RadFact (Reviewer 8v4y). The multi-stage training process lacks details on encoder comparisons, pre-training datasets, and ablation studies on sensitivity to different LLMs and configurations (Reviewer 8v4y). Reproducibility is limited due to private datasets, and comparisons to existing models are based on differing evaluation settings (Reviewers 8v4y, A4sf). Computational efficiency, cited as a contribution, is not quantitatively evaluated (Reviewers m2Vv, A4sf), and comparisons with 2D models to highlight the benefits of the 3D approach are missing (Reviewer 3EwP). Poor figure quality, inconsistent references, and limited details on experimental settings further detract from the paper's clarity and reproducibility (Reviewers A4sf, m2Vv). These issues hinder the study's clarity, applicability, and impact.

During the discussion period, the authors conducted additional experiments with clinically relevant metrics like GREEN to assess model performance (Reviewer 8v4y).They provided additional details about the pre-training datasets and clarified encoder comparisons, noting that the enhanced CTViT-V encoder was compared with the baseline. However, ablation studies on sensitivity to different LLMs and configurations were still in progress, and full results were not included in the rebuttal (Reviewer 8v4y). While the authors provided some clarification about their reliance on private datasets and their decision to use a subset of a public dataset due to time and resource constraints, the core issue of limited reproducibility remains unresolved (Reviewers 8v4y, A4sf). For computational efficiency, the authors did not provide quantitative evaluations or comparative metrics in their response (Reviewers m2Vv, A4sf).

This AC has carefully reviewed the paper, the reviewers’ initial comments, the authors’ responses, and the final feedback. While the authors provided detailed responses and addressed some concerns, significant issues remain unresolved. The reliance on private datasets raises questions about experimental design, fairness, and reproducibility, as key comparisons with existing models are based on differing evaluation settings, and public datasets were not fully utilized for validation. This limitation undermines the credibility and generalizability of the reported results. Additionally, while computational efficiency is cited as a contribution, it has not been quantitatively evaluated, leaving the claimed advantages unsubstantiated. The multi-stage training process still lacks sufficient detail, with incomplete ablation studies and limited comparisons to 2D models, which are necessary to highlight the benefits of the proposed 3D approach. The authors partially addressed the evaluation metrics concern by providing initial GREEN metric results, but their response lacked thorough validation to fully complement or replace standard NLG metrics like BLEU and ROUGE.

**Additional Comments On Reviewer Discussion:**

During the discussion period, the authors conducted additional experiments with clinically relevant metrics like GREEN to assess model performance (Reviewer 8v4y).They provided additional details about the pre-training datasets and clarified encoder comparisons, noting that the enhanced CTViT-V encoder was compared with the baseline. However, ablation studies on sensitivity to different LLMs and configurations were still in progress, and full results were not included in the rebuttal (Reviewer 8v4y). While the authors provided some clarification about their reliance on private datasets and their decision to use a subset of a public dataset due to time and resource constraints, the core issue of limited reproducibility remains unresolved (Reviewers 8v4y, A4sf). For computational efficiency, the authors did not provide quantitative evaluations or comparative metrics in their response (Reviewers m2Vv, A4sf).

---

### Decision · Program_Chairs · 2025-01-22

Reject